# Spatial pattern evaluation of a calibrated national hydrological model – a remote sensing based diagnostic approach

Gorka Mendiguren [1], Julian Koch [1], Simon Stisen [1]

[1]Department of hydrology, Geological Survey of Denmark and Greenland, Copenhagen, Denmark.

*Correspondence to*: Gorka Mendiguren (gmg@geus.dk)

**Abstract.** Distributed hydrological models are traditionally evaluated against discharge stations, emphasizing the temporal and neglecting the spatial component of a model. The present study widens the traditional paradigm by highlighting spatial patterns of evapotranspiration (ET), a key variable at the land-atmosphere interface, obtained from two different approaches at the national scale of Denmark. The first approach is based on a national water resources model (DK-Model), using the MIKE-SHE model code, and the second approach utilizes a two source energy balance model (TSEB) driven mainly by satellite remote sensing data. Ideally the hydrological model simulation and remote sensing based approach should present similar spatial patterns and driving mechanism of ET. However, the spatial comparison showed that the differences are significant and indicating insufficient spatial pattern performance of the hydrological model.

The differences in spatial patterns can partly be explained by the fact that the hydrological model is configured to run in 6 domains that are calibrated independently from each other, as it is often the case for large scale multi-basin calibrations. Furthermore, the model incorporates predefined temporal dynamics of leaf area index (LAI), root depth (RD) and crop coefficient (Kc) for each land cover type. This zonal approach of model parametrization ignores the spatio-temporal complexity of the natural system. To overcome this limitation, this study features a modified version of the DK-Model in which LAI, RD, and KC are empirically derived using remote sensing data and detailed soil property maps in order to generate a higher degree of spatio-temporal variability and spatial consistency between the 6 domains. The effects of these changes are analyzed by using the empirical orthogonal functions (EOF) analysis to evaluate spatial patterns. The EOF-analysis shows that including remote sensing derived LAI, RD and KC in the distributed hydrological model adds spatial features found in the spatial pattern of remote sensing based ET.

## 1 Introduction

The application of spatially distributed hydrological models has become common practice for a wide range of water resources assessments. Such models are valuable tools to manage terrestrial water resources and to provide insights into the overall water balance as well as the internal distribution of multiple hydrological states and fluxes. The spatial predictability of these models is however severely hampered by the general lack of suitable spatial pattern oriented model evaluation

frameworks since evaluation remains focused on spatially aggregated objective functions such as discharge. As stated by Conradt et al. (2013) "In conjunction with distributed hydrological modelling spatial calibration usually means individual multi-site calibration". The neglect of a specific focus on spatial patterns in model evaluation is a paradox in the light of an increasing acknowledgement of the role of patterns in the functioning of hydrological systems (Vereecken et al., 2016).

Moreover it is against the rationale  behind developing and applying distributed models (Freeze and Harlan, 1969;Refsgaard, 1997).  If the spatial variability of a hydrological system is not of importance to the modeler it seems not worth the effort to apply a distributed model, since numerous studies indicate that equal model fidelity can be achieved with a lumped approach when evaluated solely at the catchment outlet (Stisen et al., 2011;Vansteenkiste et al., 2014)

The concept of spatial pattern comparisons in catchment hydrology was pioneered by Grayson and Blöschl (2000), who

developed the theoretical framework and terminology. Since then significant progress has been made in areas such as model code development (Clark et al., 2015;Maxwell and Kollet, 2008;Samaniego et al., 2010), remote sensing (Lettenmaier et al., 2015) and data assimilation (Zhang et al., 2016;Ridler et al., 2014). Nevertheless, explicit spatial pattern evaluation of distributed hydrological models remains a rarity.

In order to perform qualified assessments of simulated spatial patterns reliable observations are a prerequisite. For this

purpose satellite remote sensing comes into play as an independent data source with the required spatial resolution and coverage for many catchment scale applications. Satellite imagery has been used for estimation of numerous states and fluxes of interest to hydrological modelling, such as snow cover (Immerzeel et al., 2009), ground water storage change (Chen et al., 2016;Rodell et al., 2009;Sutanudjaja et al., 2013;Richey et al., 2015), soil moisture (SM) (Wanders et al., 2014), vegetation water content (Mendiguren et al., 2015), land surface temperature (LST) (Corbari et al., 2015) or actual

evapotranspiration (ET) (Guzinski et al., 2015). The conversions of the remotely sensed signal to hydrological variables is far from trivial, and usually require in situ measurements and observations for model evaluation. However, in spite of their overall uncertainty; satellite based estimates contain valuable pattern information (Mascaro et al., 2015). More precisely, we propose to utilize remote sensing data solely for the purpose of pattern validation, using bias insensitive metrics, and leave the task to validate water balance closure to the more trusted discharge observations.

Several studies have explored the obvious potential in utilizing satellite estimates for hydrological model evaluation and calibration. Some utilize time series of a basin average observation from remote sensing to guide model calibration (Rajib et al., 2016;Rientjes et al., 2013) and therefore rely heavily on the accuracy of the remote sensing estimate while neglecting the spatial pattern information. Others utilize the satellite based estimates to perform a pixel to pixel calibration of the model (Corbari and Mancini, 2014). The latter approach might explore the full information content of the observations. However,

there is a risk of a highly parameterized solution problem and possibly unrealistic spatial parameter distributions as each grid cell is parameterized independently. Such a highly parametrized approach will be weak in light of the uncertainty of the remote sensing estimates at the pixel level. Instead we advocate approaches that seek to utilize the general pattern information of remote sensing data with less focus on specific pixels/grids and the general bias.

Relatively few studies utilize the actual pattern of remote sensing estimates in distributed hydrological model evaluation. Interesting examples are (Li et al., 2009) who used remote sensing derived patterns of actual evapotranspiration, and Hendricks Franssen et al. (2008) who utilized satellite based recharge patterns to constrain the calibration of groundwater models. Immerzeel and Droogers (2008) included actual evapotranspiration estimates in the calibration of the Krishna basin
in southern India and obtained a good correlation across 115 sub basins while Githui et al. (2012) successfully applied a multi-objective calibration by combining river discharge and remotely sensed actual evapotranspiration of 59 sub basins in Victoria, Australia. Other pattern oriented model evaluations without calibration were conducted by Bertoldi et al. (2010), Wang et al. (2009) and Koch et al. (2016); the latter applied different spatial performance metrics in the evaluation of three land surface models over the continental United States based on remotely sensed land surface temperature maps.
The aim of this study is to evaluate the spatial pattern performance of the national hydrological model of Denmark (DK-Model). In order to achieve this, a secondary goal is to develop a thermal remote sensing based actual evapotranspiration (AET) dataset suitable for validation and calibration of  the large scale distributed hydrological model . The idea is to thoroughly investigate the observed differences in spatial patterns between observations and simulations in order to understanding the underlying processes that generate patterns in actual evapotranspiration. The model evaluation is based on
a diagnostic approach inspired by the study of Schuurmans et al. (2003) who utilized satellite estimates to identify conceptual model errors in a small sub basin of the MetaSWAP model in the Netherlands. This approach  aims at identifying which parts of the model parametrization generates these differences in the spatial patterns. Later, in an attempt to increase the similarity between the observed and simulated pattern of ET some new inputs and parameter distribution schemes are generated and included in a modified version of the DK-Model. The response of these modifications in the modified DK-
Model are later evaluated against both spatial patterns of evapotranspiration,   stream discharge and ground water heads. Results show  that newly gained insights can guide the development of a new parameterization scheme and calibration framework that can facilitate an improvement in the spatial model performance.

## 2 Methods

In this study two approaches are undertaken to estimate spatial patterns of evapotranspiration (ET) at national scale
(Denmark) (Fig. 1); the first is based on a Two Source Energy Balance (TSEB) driven by remote sensing data (section 2.1); and the second is based on a distributed hydrological model (DK-Model) (described in section 2.2). We acknowledge that both approaches are subject to uncertainties; however, the aim of this study is to evaluate the dominating spatial pattern across Denmark and to gain insights into which processes and variables generate these patterns. The evaluation will focus on the spatial pattern itself by neglecting differences in the absolute values of evapotranspiration.
In the last step of this study the current version of the DK-Model is modified by replacing the original root depth (RD), crop coefficient (Kc) and leaf area index (LAI) based on lookup tables and a land cover map by remote sensing based data which features detailed spatio-temporal information

## 2.1 TSEB setup

### 2.1.1 TSEB theory

The Two Source Energy Balance Model (TSEB) proposed by Norman et al. (1995) is used to retrieve mean monthly maps of

ET across Denmark for the period 2001-2014. Although several global remote sensing based data sets of actual evapotranspiration, such as MODIS ET (Mu et al., 2007) and GLEAM (Miralles et al., 2011), are available, the TSEB model was selected as the most appropriate remote sensing algorithm for the current study. The main reason is that TSEB is mainly driven by land surface temperature, which is a key indicator of evaporative state at the surface (Kalma et al., 2008). This makes LST based ET algorithms more appropriate than purely vegetation based algorithms such as (Mu et al., 2007). Other

models, such as GLEAM (also not including LST), can be considered a fusion of models and remote sensing data (e.g. including a soil moisture model and plant water stress model) and as such become difficult to regard as an observation.

In our study we have incorporated the code which is provided by the pyTSEB package (https://github.com/hectornieto/pyTSEB last accessed 30/01/2017). The applied model is a two layer model that treats soil and vegetation separately and estimates fluxes on the basis of remotely sensed LST, albedo and vegetation parameters in

combination with the climate variables air temperature ($T_{Air}$), wind speed, shortwave and longwave radiation. As presented in Norman et al. (1995), and presented here in a simplified way, the model is based on the energy balance equation:

$$LE[Wm^{-2}] = R_n - H - G = LE_C + LE_S = (R_{nC} - H_C) + (R_{nS} - H_S - G_S) \tag{1}$$

Where H is the sensible heat flux, G is the ground heat flux, $G_S = 0.35R_{ns}$, $R_n$ is the net radiation and LE is the latent heat flux all in Wm$^{-2}$, with subscripts C and S represents canopy and soil respectively.

The sensible heat flux (H) is calculated as:

$$H[Wm^{-2}] = H_C + H_S = \rho C_P \left[\frac{T_C - T_A}{R_A}\right] + \rho C_P \left[\frac{T_S - T_A}{R_A + R_S}\right] \tag{2}$$

Where $H_C$ and $H_S$ are the sensible heat flux for the canopy and soil respectively, $T_C$ and $T_S$ are the canopy and soil temperatures (K), $\rho C_P$ is the volumetric heat capacity of air (Jm$^{-2}$s$^{-1}$), $R_S$ is the resistance to heat (s m$^{-1}$) flow in the boundary layer above the soil surface and $R_A$ is the aerodynamic resistance (s m$^{-1}$) expressed as:

$$R_S[sm^{-1}] = \frac{1}{c(T_S - T_C)^{1/3} + bu_S} \tag{3}$$

with c = 0.0025 and b = 0.012 and $u_S$ being the wind speed at a height above the soil surface where the effect of the soil surface roughness is minimal.

$$R_A[sm^{-1}] = \frac{\left[ln\left(\frac{z_U - d}{z_M}\right) - \Psi_M\right]\left[ln\left(\frac{z_T - d}{z_M}\right) - \Psi_H\right]}{0.16U}, \tag{4}$$

where $z_U$ and $z_T$ are the height of the wind speed (U) and air temperature in meters, $\Psi_M$ and $\Psi_H$ are the adiabatic correction

factors for momentum and heat, $d$ is the displacement height ($d \approx 0.65\, h_c$, and $h_c$ is the height of the canopy (m)), $z_M$ is the displacement height for momentum ($z_M \approx h_c/8$).

The unknowns of Eq. 2 are $T_C$ and $T_S$, which are related to the observed directional LST($\theta$) by:

$$LST(\theta)[K] \approx \left( f(\theta)T_C^4 + (1 - f(\theta)T_S^4) \right)^{1/4} \tag{5}$$

Where $f$ is the fraction of view of the radiometer occupied by vegetation, and $\theta$ is the viewing angle.

$$f(\theta) = 1 - exp\left( \frac{-0.5LAI}{cos\theta} \right) \tag{6}$$

In order to solve for $T_C$ and $T_S$ a second expression is required, which in the TSEB approach originates from the Priestley-Taylor approximation (Priestley and Taylor, 1972) used here for an initial estimate of the latent heat flux for the green part of the canopy, assuming a $\alpha_{PT} = 1.26$, corresponding to potential transpiration. The initial transpiration is given by the equation:

$$LE_C[Wm^{-2}] = \alpha_{PT} f_g \frac{S}{S+\gamma} R_{nC}, \tag{7}$$

Where $f_g$ represents the fraction of LAI that is green, $S$ is the slope of the saturation vapour versus temperature curve and $\gamma$ is

the psychrometric constant and where the net radiation in the canopy ($R_{nc}$) is calculated based on the separate components of direct and diffuse shortwave and longwave radiation for the canopy as described by (Campbell and Norman, 1998;Kustas and Norman, 1999) accounting for albedo, extinction coefficient, LAI.

By combining Eq. 1 and 7, $H_C$ can be derived.

$$H_C[Wm^{-2}] = R_{nC} - LE_C = R_{nC}\left( 1 - \alpha_{PT} f_g \frac{S}{S+\gamma} \right) \tag{8}$$

After calculating the initial $H_C$ (Eq. 8) and the resulting $T_C$ (Eq. 2), Eq.5 is used to calculate first $T_S$, then the sensible and latent heat fluxes for soil ($H_S$ and $LE_S$) based on Eq. 2 and 1. The initial estimates of fluxes do however not guarantee that the energy balances are satisfied. If the calculated $LE_S$ is less than zero, TSEB internally modifies the value of $\alpha_{PT}$. When $LE_S < 0$ is encountered, the canopy is assumed to be stressed and $\alpha_{PT}$ (in Eq. 7) is iteratively reduced until solutions with $LE_S > 0$ are obtained. The model iteration continues until the energy balance equations are satisfied for both soil and canopy. The

readers are referred to Norman et al. (1995) and the github pyTSEB package to find a full description of the model.

### 2.1.2 Derived remote sensing inputs

Several inputs to TSEB are directly obtained from the Moderate Resolution Imaging Spectroradiometer (MODIS) sensor all at 1 km spatial resolution; day time LST and day time VZA (View Zenith Angle) obtained from MOD11A1 and MYD11A1 products on board of on TERRA and AQUA satellites respectively. The decision of whether to use LST from TERRA or

AQUA is based on the percentage of high quality pixels available covering Denmark in each scene. The quality flags included in the products are used to select only those pixels with the best observation possible, no cloud present, no cloud shadow etc. In addition only satellite observations obtained between 11:00 to 13:00 local solar time are utilized to ensure minimum effect of acquisition time.

This study focuses on the growing season from April to September. From a water resources perspective the spatial patterns

of ET are regarded as irrelevant for the remaining months of the year due to energy limited conditions and very low potential evapotranspiration. First the Nadir BRDF (Bidirectional Reflectance Distribution Function) Adjusted Reflectance (NBAR)

from the MCD43B4 product for that time period was used to calculate the NDVI (Rouse et al., 1973) using the following equation (Eq. 9):

$$NDVI[-] = \frac{B_2 - B_1}{B_2 + B_1},$$
(7)(9)

where B1 and B2 is the reflectance from bands 1 and 2 from MODIS (645.5 nm and 856.5 .nm respectively).

Later, the Savitzky-Golay filter (Savitzky and Golay, 1964) available in the TIMESAT code (Jönsson and Eklundh, 2002;Jönsson and Eklundh, 2004) is selected to smooth the NDVI time series as it preserves the maximum and minimum values of the original dataset and guarantees consistency of time series. In situ measurements of LAI are usually expensive and time consuming, therefore reference LAI was based on the tables used for the Danish National Water Resources model (Stisen et al., 2012) which are based on previous works of Refsgaard et al. (2011). Boegh et al. (2004) derived LAI

variability in Denmark using NDVI with an exponential function. In this study a similar approach where coefficients are adjusted to match the input LAI data used in the National Water Resources model resulting in Eq. 10:

$$LAI\left[\frac{m^2}{m^2}\right] = \alpha \cdot e^{\beta \cdot NDVI}$$
(10)

Where $\alpha$ and $\beta$ are specific parameters for this study case and that is adjusted to maximize the fit between the calculated NDVI and the reference LAI from the DK-Model, leading to Eq. 11:

$$LAI\left[\frac{m^2}{m^2}\right] = 0.0633 \cdot e^{5.524 \cdot NDVI}$$
(11)

Another parameter that is needed to run the TSEB is the Vegetation Height (VH). This is derived assuming a simple linear regression as in Stisen et al. (2011) following Eq. 12 :

$$VH[m] = 0.1 \cdot Height_{Max} + 0.9 \cdot Height_{Max} \cdot \left(\frac{LAI_i}{LAI_{i,Max}}\right)$$
(12)

Where $Height_{Max}$ is a value that changes for each land use class , $LAI_{i,Max}$ indicates the maximum LAI value for a particular

pixel. This relationship is applied on all land use classes except forest, which is set to a year round constant VH of 15m. The input albedo data was obtained from the MODIS 8-day MCD43B3 product using only good quality pixel according to the quality flag (MCD43B2). In order to further reduce noise, mean albedo maps are generated by creating 46 mean maps (at 8 day intervals) using all the scenes available for each 8-day interval across different years. The albedo mean maps for each 8 day period are later used to calculate the net radiation in TSEB.

Climate forcing data to run the TSEB are obtained from the ERA-Interim reanalysis data set (Berrisford et al., 2011;Dee et al., 2011) provided by the European Centre for Medium-Range Weather Forecast (ECMWF). We have incorporated 10 m horizontal wind speed, 2 m air temperature, surface solar radiation and longwave incoming radiation.

Fraction of Green vegetation was derived from LAI following Eq. 13:

$$Fg_i[-] = \frac{LAI_i}{LAI_{MaxClass}}$$
(13)

Where $Fg_i$ indicates the Fraction of green for a certain pixel $i$ , $LAI_i$ indicates the LAI value for a pixel i  and $LAI_{MaxClass}$ is the maximum LAI value for an specific land cover type. The approach is a simplified form of the vegetation index based method by Gutman and Ignatov (1998). This equation was applied to needle leaf forest land cover type.

For the other land cover types (crops, grasslands, deciduous forest etc.) Eq.13 was modified adding another term. These land cover types show a stronger seasonality and a clear distinction between a greening phase and a senescence phase. In order to represent the strong difference in fraction of green vegetation between the period before and after senescence we introduced a different equation for the period between crop emergence and senescence, where we assigned higher values of Fg to non-needle leaf forest land covers, Fig. 2.   For these types of vegetation Fg will be allowed to increase rapidly just after crop emergence by substituting Eq. 13 by Eq. 14.

$$Fg_i[-] = \frac{LAI_{i,max}}{LAI_{MaxClass}} \cdot (1 - e^{(-2 \cdot LAI_i)}) \tag{14}$$

Where $LAI_{i,max}$ indicates the maximum LAI value for a pixel i.

This substitution is only conducted during part of the phenological year, more specifically for the period starting at the green-up date, corresponding to the point defined by an increase in 20% in LAI compared to the winter low ($LAI_{i,Min}$) (Cong et al., 2012) and continuing until the time at which LAI reaches its maximum ($LAI_{i,max}$) (Fig. 2 ). This approach will mediate the shortcomings of the vegetation index based methods, which has been shown to underestimate fraction of green during the greening phase while corresponding well to field observations during senescence (Guzinski et al., 2013).

Finally, instantaneous estimates of ET are converted to daily ET values based on the assumption of a constant value of the evaporative fraction throughout the day (Sugita and Brutsaert, 1991;Brutsaert and Sugita, 1992) following  Eq. 15 :

$$dET[Wm^{-2}] = EF \cdot dRn \tag{15}$$

where dET represents the daily ET  and EF is the Evaporative Fraction and dRn represents the daily net radiation and where EF is calculated as in Eq.16:

$$EF[-] = ET/Rn \tag{16}$$

Where ET is the instantaneous actual evapotranspiration and Rn is the instantaneous Net Radiation at the same acquisition time. The assumption of constant EF over the course of a day is often not completely true and also affects the estimates of daily ET (Gentine et al., 2007) but in the current application is not crucial since only the spatial pattern of the remote sensing estimates is utilized.

Later, the daily maps are aggregated to monthly mean maps by using only those days in which the national coverage exceeded 50%. The final monthly mean maps comprises all available ET estimates for a given month across all years (2001-2014) resulting in just six climatological maps (April-Sept). This temporal aggregation process is conducted in the same way with simulations from the DK-Model considering only the same cloud free pixels/grids as from the TSEB estimate and ensuring the comparability of the maps.

### 2.1.3 Optimization of vegetation parameters

Data from three eddy covariance (EC) flux towers is used as a reference to perform a sensitivity analysis and calibration of some of the vegetation parameters of the TSEB. A detailed description of the instrumentation and data processing of each of the flux sites can be found in Ringgaard et al. (2011). Latent heat (LE) or evapotranspiration measurements are obtained at a

frequency of 30 min and the mean value of the observations from 11:00 to 13:00 were used as reference in the calibration of TSEB for the cloud free days where the TSEB estimates are available. The observed eddy covariance data are subject to energy balance closure problems, typically in the order of 20-25%, which is usual for this type of measurements (Hendricks Franssen et al., 2008).

The evaluation of the TSEB was conducted using as reference the data of the EC systems from the 3 different land cover

types that were corrected for the energy closure using the Bowen ratio approach (Bowen, 1926). The associated uncertainty is the span between all closure error being assigned to either sensible heat or to latent heat.

In order to identify the input and parameters that assert the main control of the TSEB model a sensitivity analysis is conducted with the help of PEST (http://www.pesthomepage.org/Home.php), a model-independent parameter estimation and uncertainty analysis tool. PEST evaluates the sensitivity of each parameter by perturbing the value of each of the parameters

one at a time and subsequently analyse the response of the performed perturbation with respect to a change in model performance. At last, the sensitivities are normalized using the most sensitive parameter as reference.

Later an optimization of selected vegetation parameters is performed. The objective function is set to reduce the differences in mean monthly ET estimates at three EC measurements sites that represent the three main land cover types in Denmark (Agriculture, Forest and Meadow).

Only 4 parameters are calibrated using PEST. These parameters are: Priestley Taylor parameter for forested areas ($PT_{Forest}$), and the $LAI_{MaxClass}$ for the three land covers used to estimate the Fg ($LAI_{Max}^{Agriculture}$, $LAI_{Max}^{Forest}$, $LAI_{Max}^{Meadow}$).

### 2.2 Hydrological model

The National Water Resources Model (DK-Model) (Højberg et al., 2013;Stisen et al., 2012) was developed at the Geological Survey of Denmark and Greenland in 1996 and updated several times (Henriksen et al., 2003;Højberg et al., 2013). The

model is constructed within the hydrological model system MIKE-SHE (Abbott et al., 1986). The model works at 500 m resolution and due to computational efficiency and differences in geology the DK-Model was divided into 7 different domains that cover the entire country; however in this study only 6 of the 7 domains were selected covering approximately 98% of the country with and extension of 42.087 km$^2$. Domains used are presented in Fig. 1. The model is based on a full 3-D finite difference groundwater module that is connected to a simplified two–layer unsaturated zone module (Yan and

Smith, 1994). Furthermore, the model was previously calibrated using 191 discharge stations and approximately 17.500 data entries of ground water head (GWH) (Stisen et al., 2012 ). The DK-Model has been extensively used in different applications with different objectives; assessment of climatic change (Karlsson et al., 2016) , water resources management (Henriksen et

al., 2008), large scale nitrogen modelling (Windolf et al., 2011;Hansen et al., 2009;van der Keur et al., 2008)  highlighting the importance of the spatial component of the model and its reliability.

### 2.2.1 Remote sensing derived hydrological model input data

Part of the current study is to identify model inadequacies and test possible directions of model parametrization improvements. In an attempt to improve the initial DK-Model, spatially and temporally distributed root depth (RD) maps are generated using remote sensing data using a vegetation index based approach (Koch et al. (2017)) where RD is calculated by Eq. 17:

$$RD_i[m] = RD_{max} \frac{NDVI_i}{NDVI_{max}} \tag{17}$$

where $RD_i$ is root depth in pixel $i$, $NDVI_i$ is the NDVI in that cell, and $NDVI_{max}$ is the and $RD_{max}$ indicate the maximum values at $i$ in meters. This equation was used for forested land cover types, poorly vegetated areas and urban areas. RD can be considered an effective parameter in the DK-Model which partly compensates for the lack of a specific vegetation component in the evapotranspiration calculations, and therefore variability in LAI and phenology. For instance, Eq. 17 equips forest cells with a temporal varying RD which is contrary to our physical understanding, but it compensates for mixed land-use cells and undergrowth. RD of croplands and grasslands, denoted as $RD_{Agri}$, was estimated by implementing some modifications to Eq. 17. For Danish agricultural land covers, the effective maximum root depth ($RD_{max}$) is known to be lower for the very sandy soils in western Denmark (Refsgaard et al., 2011;Breuning Madsen and Platou, 1983). This dependency of maximum root depth on soil type is accounted for by a linear relation between the clay fraction (CF) in the soil and $RD_{max}$. This relation is then included as a substitute of $RD_{max}$. In order to allow RD to reach zero for croplands, the second term in Eq. 17 is normalized by including $NDVI_{min}$ in the Eq. 18,

$$RD_{(agri)i}[m] = [(\alpha_{RD} \cdot CF_i) + \beta_{RD}] \cdot \frac{NDVI_i - NDVI_{min}}{NDVI_{max} - NDVI_{min}} \tag{18}$$

Where $CF_i$ indicates the clay fraction at pixel $i$, $NDVI_i$ indicates the NDVI value of pixel $i$, and $NDVI_{Min}$ and $NDVI_{Max}$ represent the minimum and maximum values of NDVI for the same pixels.

The constants $\alpha_{RD}$ and $\beta_{RD}$ are considered calibration parameters that should be tuned to the best overall water balance and spatial pattern in AET. For the initial run of the modified DK-Model, values of $\alpha_{RD} = 12$ and $\beta_{RD} = 0.2$ are assigned. These values are derived by matching the average root depth across all grids obtained through Eq. 18 with  the corresponding average root depth of the original DK-Model.

In addition, the crop coefficient (Kc), which is a correction factor for the reference evapotranspiration ($ET_{ref}$) was recalculated. The $ET_{ref}$, describes the climatologically based actual evapotranspiration for the reference crop (a short grass without water stress) and is here provided at a coarse spatial resolution of 20 km. The Kc-value accounts for the difference between a given crop or land surface and the reference crop by scaling the $ET_{ref}$ to the potential evapotranspiration used in the hydrological model. In the original DK-model setup, Kc was based on lookup tables for different land covers.  In the

modified parameterisation, Kc is derived from remotely sensed LAI using the approach presented in Allen et al. (1998) and used by Stisen et al. (2008):

$$\text{Kc}[-] = \text{Kc}_{c,min} + \left(\text{Kc}_{c,max} - \text{Kc}_{c,min}\right) \cdot \left(1 - e^{(-0.7 \cdot \text{LAI})}\right) = 0.95 + 0.2 * (1 - e^{(-0.7 * \text{LAI})}) \tag{19}$$

Where the $\text{Kc}_{min}$ and $\text{Kc}_{max}$ are set to 0.95 and 1.15 respctively.

## 2.3 Spatial pattern analysis; Empirical Orthogonal Functions (EOF)

The Empirical-Orthogonal-Functions (EOF) analysis is a statistical technique commonly used to evaluate large spatio-temporal datasets of hydrological states and fluxes (Mascaro et al., 2015;Perry and Niemann, 2007;Graf et al., 2014). It can be applied on either the spatial or the temporal anomalies and this should be reflected by how the data matrix is prepared. Most commonly, and also applicable for our study, the EOF analysis is applied focusing on the spatial variability. For this purpose the rows in the data matrix represent the locations and the columns, each having a sum of zero, the timesteps. When being applied, the EOF analysis decomposes the variability of the spatio-temporal data matrix in two main components. First, a set of orthogonal spatial patterns (EOFs) which are time invariant and define statistically significant patterns of covariation. Second, a set of loadings that are time variant and specifying the importance of each EOF over time. Graf et al. (2014) and Perry and Niemann (2007) described briefly the mathematical background of the EOF analysis. Most commonly, the EOF analysis is applied on either observational or modelled datasets, but recent applications stressed it's usability as a tool for spatial validation of distributed hydrological models at catchment scale (Fang et al., 2015;Koch et al., 2016 ). Koch et al. (2015) suggested performing a joint EOF analysis on an integral data matrix that contains both, observed and simulated data which are concatenated along the temporal axis doubling the number of columns. In this way, the resulting EOF maps honour the spatio-temporal variability of both datasets and the weighted difference between the loadings at specific times can be utilized to derive a quantitative pattern similarity score. The weighting is necessary, because the amount of covariation that lies in each EOF differs, where the first EOF is oriented in the direction of maximum covariation. Therefore, the EOF based similarity score (SEOF) between an observed and a predicted ET map at time x can be formulated as Eq. 20:

$$S_{EOF}^{x} = \sum_{i=1}^{n} w_i \left| \left( \text{load}_i^{simx} - \text{load}_i^{obsx} \right) \right| \tag{20}$$

where $w_i$, the covariation contribution of the i'th EOF, is multiplied with the absolute difference between the simulated loading (load$^{sim}$) and the observed loading (load$^{obs}$) of the i'th EOF at time x.

The EOF analysis applied in this study evaluated the differences in spatial patterns between the DK model outputs in the original configuration and a modified version where three inputs (RD, LAI and Kc) of the model were replaced by those derived from remote sensing data.

# 3 Results and discussion

The results and discussion are presented in two sections; the first focuses on the sensitivity analysis and parameter optimization of the TSEB model and the second features the spatial pattern evaluation of the DK-Model using the maps obtained from the TSEB model.

## 3.1 Sensitivity analysis and TSEB calibration

The normalized sensitivity values of the 23 incorporated variables and parameters are illustrated in Fig. 3. The results are presented in three groups depending on the group they belong to: remote sensing data, forcing data and vegetation parameters.

The results show that the most sensitive variable for the estimation of AET is LST. Interpreting the sensitivity values for each group individually stress that, for the remote sensing input, parameters that are directly related to LST such as emissivity of vegetation (EmissV) and soil (EmissS) are characterized by a high sensitivity as well. The next group, forcing data, exhibited high sensitivity for all variables, except for wind speed. Overall Air temperature (TempAir) is the most sensitive forcing variable. These results indicate that the algorithm is largely controlled by the LST, LAI and climate forcing data. This finding is considered ideal, since the actual parameters of the algorithm do not dominate the final spatial pattern. In general, the remote sensing and forcing data inputs can be considered observations which are not subject to calibration.

The sensitivity analysis was utilized for illustrating the main controlling variables of the TSEB algorithm, not for selection of calibration parameters. Subjectively, four vegetation related parameters were selected for calibration to optimize the match to each of the three land cover types. First, the $PT_{Forest}$ parameter was selected because the Priestly-taylor coefficient of forests is believed to be below the standard value of 1.26 assigned for agriculture and meadow (Komatsu, 2005). Secondly, the $LAI_{Maxclass}$ values ($LAI_{Max}^{Agriculture}$, $LAI_{Max}^{Forest}$, $LAI_{Max}^{Meadow}$.) which control the fraction of green vegetation (Fg) through Eq. 13 and 14 are selected for calibration.

The results of monthly ET estimates are presented for all three sites in Fig. 4. The bars on the observed values indicate the uncertainty associated with the energy balance closure issues where the upper bound of the uncertainty bar represent the situation in which the residual energy is assigned to the latent heat (LE), whilst the lower bound represent the opposite situation in which all the residual energy is assigned to the sensible heat (H).

Generally the estimated ET values agree well with the EC-measurements especially considering the uncertainties associated with energy balance closure and the spatial scale mismatch between the EC footprint and the remote sensing estimations. In order to minimize the effect of scale issues the EC values of ET at the three sites are compared to the average ET of the surrounding pixels estimated by TSEB. For this comparison, pixels that are considered as purely representative of the specific land cover type and therefore not contaminated by other land cover types are used. The selection of the pixels is carried out manually with the help of a high resolution image of the study area.

The comparison is meant as an illustration of the ability of the TSEB to describe the general annual variation and differentiate between land covers. The main aim of the TSEB application is to get robust national maps of growing season ET and the results show agreement on both, the seasonal variation and absolute levels of ET. On the other hand the separation between land covers is somewhat harder to evaluate because all three sites exhibit a similar level of ET.

ET in the forested areas remains mostly constant during the growing season with a tendency to increase at the end. Agricultural areas on the other hand presented much higher variability with a rapid increase at the beginning of the growing season (May-June) and a decrease at the end (August-September). The Wetland shows a similar shape as the forest but with slightly higher ET values, and presents a big increase in the month of August that is not capture in the TSEB.

Mean monthly maps of ET are generated from daily TSEB estimations across all years to ensure consistent spatial patterns
for robust spatial model evaluation with the aim to evaluate and improve the model performance. Such an improvement can be facilitated through optimal parametrization, and we therefore focus on the consistent spatial patterns rather than the temporal dynamics of ET variability. This is also reflected in the way the TSEB ET estimates are evaluated.

Results indicate that the TSEB ET estimates are within the measurement uncertainty of the EC at the three stations. The only pronounced disagreement is observed in the wetland during the month of August. Ringgaard et al. (2011) showed how the
water level of the Skjern River raised during that time of the year and therefore increasing the values of ET in the EC measurements which is located at the bank of the river.

## 3.2 Spatial patterns

The mean monthly maps of cloud free ET generated with the TSEB model and DK-Model are presented in Fig. 5 and Fig. 6. For a better visualization of the spatial patterns the maps were normalized in this case by dividing each map by the mean
value of the map itself. The TSEB ET (Fig. 5) exhibits a clear difference between Eastern and Western Denmark with lower ET values in the sandy Western Denmark especially in the peak of the growing season (May-June). The clear E-W pattern identified by the TSEB model is remarkable considering that it is opposite the general precipitation gradient (Fig 1). This highlights the strong influence of soil properties on the ET pattern across Denmark. Another feature is that forest areas have lower ET for the selected cloud free days where canopy interception is not included.

Regarding the results from the DK model (Fig. 6) it can be observed that the E-W trend is not noticeable in the maps and the difference between forest and agriculture is less distinguishable. Moreover the effects of the zonal calibration are causing differences in model domain 2 (Fig. 1) which have much higher ET in comparison to the other domains, especially in May and June. From Figs. 5 and 6 it can be extracted that there is almost no resemblance between the spatial patterns identified in the TSEB ET and the DK-Model simulations on the national scale. This seems substantial since the model has been
calibrated extensively. However, the applied discharge based calibration is dominated by the winter peak runoff, which conveys little information with respect to the spatial patterns of summer ET. This finding actually highlights the need for spatial pattern evaluation of distributed hydrological models since traditional discharge and groundwater head calibration does not necessarily lead to reasonable ET patterns.

In a first attempt to improve the simulated spatial patterns of the DK-Model, new parameterizations of RD, LAI and Kc are prepared based on fully distributed remote sensing and soil data as explained in section 2.2.1. These contain a higher degree of spatio-temporal detail than the original model input based on predefined tables from Refsgaard et al. (2011) and should reflect distributions that are more realistic and spatially consistent. Fig. 7 shows the modified DK-Model mean maps of ET. The patterns of these maps are more similar to those observed with the TSEB in which the E-W pattern is quite evident, although this pattern seems to be exaggerated.

Fig. 8 shows the spatial differences in growing season average ET between the DK-Model in its original (r = 0.07) configuration and the modified version (r = 0.33) based on remote sensing input. It is important to highlight at this point that the modified DK-Model is not recalibrated with the new inputs as this goes beyond the scope of this study. A recalibration may modify the water balance in comparison to the original setup. However the performed modifications show some relevant features; the most noticeable visual improvement is the much larger gradient in the East-West pattern obtained in the modified DK-Model, which emphasizes the more distinct resemblance to the pattern, estimated with TSEB which also translated in an improvement in the Pearson coefficient from 0.07 to 0.33. Visually this improvement can be attributed to a more clear East-West pattern and smoother transition in the values of domains 1,2 and 3 in the modified version compared to the original DK-Model. Similarly, Fig. 9 underlines that the changes in the original setup of the DK-Model and the modified version are large when compared using scatter plots using the mean normalized map of TSEB as reference. Even the dispersion in the scatter plots is large, the results reveal an improvement in the Pearson correlation coefficient and also the points move closer to the 1:1 line.

To analyse the driving mechanism behind the "observed" TSEB patterns and the simulated DK-Model patterns the clay fraction used as input to the root depth calculation of the modified DK-Model, the observed average LST input to the TSEB model and the growing season average LAI are illustrated in Fig. 10. The similarities between LST and clay fraction maps with TSEB are quite evident (r= -0.50 and r=0.44 respectively) whilst the similarity with LAI is low (r = -0.15).

These maps reveal interesting findings; first the presence of the East-West pattern in the clay fraction map coincides visually quite well with the TSEB model mean outputs (Fig. 8) in spite of the fact that no soil information has been included in the TSEB ET estimation. This indicates that the general perception, of lower ET for the sandy soils in the West due to soil moisture stress in the summer period, is captured well by the TSEB ET. The East-West pattern is not captured by the DK-Model simulation even though the model is based on soil type information on field capacity and wilting point. On the other hand the modified DK-Model captures much more of the East-West pattern because the clay fraction information is utilized to stretch the root depth distribution. Moreover, the TSEB pattern is mainly controlled by the LST input combined with a fine scale variability introduced by the LAI patterns (Fig. 10).

The EOF analysis (Fig. 11) extracts the spatio-temporal similarities and dissimilarities between the two different DK-Model configurations. The analysis is based on monthly mean maps generated using the daily simulations. The integral data matrix, containing both DK-Models, has 38188 rows reflecting the number of cells and 144 columns containing 144/2 observed and

simulated maps concatenated along the temporal axis. Only the first three EOFs which, in combination, explain 71% of the total variance are presented. The first EOF captures 45.2% variance and the EOF loadings present very small differences and are equipped with positive sign throughout the entire period. Hence it can be interpreted that EOF 1 addresses the major similarities between the two model configurations. The EOF1 map captured the component of the ET pattern which is driven

by the soil properties, as it relates nicely with the mapped clay content in Fig. 9. The loadings of the second EOF in combination with its map add 15.7% to the explained variance. The apparent disagreement in values and sign between the loadings stressed that EOF2 captures the major dissimilarities between the two model configurations. The EOF2 pattern resembles the one found in EOF1, however it was characterized by less contrast and overall, it represents the added spatial detail of RD which is defined as a function of clay content and vegetation. The evident E-W trend is strongest in the first

three months of the growing season and afterwards the loadings drop to close to zero for the modified DK-Model. The third EOF explains around 10% of variance and further records dissimilarities between the two models. Examining the loadings stresses that the modified DK-Model plays a minor role in EOF3, as loadings are close to zero. However the first three months of the original DK-Model seems well represented and the map underlines the granularity of the original setup, which is strongly driven by the discrete land-use map. Also, the model boundary between area 5 and 6 appears in EOF3, which was

caused by the zonal calibration of RD in the original DK-Model. The overall similarity scores derived by the EOF analysis presented the maximum value for a pattern comparison in June (≈0.11) and the minimum corresponded to a day in April (≈0.02).

The results highlight a soil properties driven spatial pattern which is expected due to larger water holding capacity in clay dominated areas. This relationship is clearly evident in the TSEB data, although soil data do not drive the TSEB algorithm,

but this information is embedded in the LST as LST can be used to map soil textures (Wang et al., 2015). In contrast the original DK-Model includes soil type information, but clearly the soil parametrization does not have sufficient effect on the simulated patterns of ET. The spatial patterns can probably be improved through calibration, by increasing the contrast in soil parametrizations or by modifying the model formulations on the soil stress function. In the current study, a new root depth parametrization is applied where the spatio-temporal variation in the effective root depth is estimated based on a

combination of the clay fraction map and remotely sensed NDVI time series. The simulated ET maps resulting from the new remote sensing based DK-Model (including root depth Kc and LAI) are clearly much more similar to the TSEB (Fig. 10) although significant differences still occur in some regions. In order to achieve a better performance the transfer-function of the root depth and Kc parametrizations have to be calibrated against the spatial patterns of the TSEB (in combination with discharge and groundwater head). Unfortunately, re-calibration of the National DK-Model goes beyond the scope of the

current study, since a single model run of the entire DK-Model requires around 40 hours (wall-clock time), but re-calibration will be part of future improvements of the model. This will ensure both spatially consistent parameterizations by utilizing transfer functions inspired by the parameterization scheme of the mesoscale Hydrologic Model (mHM) (Samaniego et al., 2010) and an optimal trade-off between discharge, groundwater head and spatial patterns of ET.

The applied EOF analysis identifies the spatio-temporal similarities and dissimilarities between the two DK-Model configurations. It allows pointing out driving mechanisms behind the simulated spatial patterns, such as the effect of the effective RD in the modified DK-Model in EOF2 or the sharp boundary of simulated ET caused by the zonal calibration of the original DK-Model (EOF3). These findings strengthened the EOF analysis as a suitable tool to meaningfully compare

spatial patterns and to diagnose spatial model deficiencies. Recently, the proposed approach was applied by Koch et al. (2017) and Ruiz-Pérez et al. (2016) in a spatial sensitivity analysis and in a spatial pattern oriented model calibration, respectively. In the future the EOF analysis will be considered as a metric to re-calibrate the DK-Model with focus on spatial patterns of ET.

### 3.2.1 Key differences between the models

The two different approaches to retrieve ET are compared based on the idea that both models, TSEB and the DK-Model should present similar spatial pattern of ET. Results showed that the differences in the outputs were noticeable. These differences can be divided in three groups.

- Differences due to model setup: The DK-Model is an aggregate of 6 domains. Each of these domains is calibrated individually, which leads to inconsistent spatial distributions of hydrological properties across domains. This increases

the accuracy and performance of the model when evaluated only using discharge stations and ground water heads, but ignores the spatial component as it is an aggregated evaluation. On the contrary, when the ET maps are obtained with TSEB these problems are not present as all the study area is treated the same way and with the same parameterization.

- Spatial differences due to the land cover parameterizations are also important and are clearly evident in the case of the TSEB maps when comparing forest and agriculture areas. In this study three different EC datasets are used to calibrate

the vegetation parameters and these sites are assumed to be representative of each land cover at a national scale. In some cases this assumption might not be adequate as soil type/ forest type and other variables might affect the plant response to ET. The TSEB was adjusted to show this pattern which might in some cases be overestimated and therefore enhancing the contrast of the TSEB between two land cover types.

- Differences due to the models: Estimations of ET in the DK-Model are mainly driven by precipitation, root depth and

soil properties and represent a residual in the water balance equation. On the other hand the TSEB relies mostly on forcing data and LST to estimate ET as a residual in the energy balance equation and does not take into account any soil information or rainfall. The similar features found between the mean annual maps of TSEB and clay fraction may indicate that the LST, and thereby TSEB, is sensitive to some soil properties. On the other hand these similarities between LST and soil property patterns can also be explained by the fact that areas with sandy soils and low clay

fraction are coincident with areas with lower agricultural production and higher risk of summer drought and vegetation under soil water stress.

### 3.3 Stream discharge and groundwater head performance

Besides comparing the spatial patterns of the original and modified DK-Model, the stream discharge and groundwater head performance is also compared. In this comparison it is important to acknowledge that the original DK- model has been calibrated against these variables, whereas the evaluation of the modified DK-Model has to be considered as a validation.

Results showing the annual and summer (Jun-Jul-Aug) runoff volume error (WBE) as well as NSE (Nash-Sutcliffe Efficiency) for 181 discharge stations are presented in Fig. 12. The first noticeable thing that can be concluded is that the average water balance error changes from a slight overestimation to a moderate underestimation (Median WBE changes from -5.5 % to 5.5% for the original and modified models respectively). Regarding, the summer water balance which is expected to be influenced the most by the model modifications; the picture is similar although the performance get worse

with a larger positive bias. The NSE showed a decrease in performance, from NSE= 0.72 in the original DK model to NSE=0.67 in the modified version.

Ground water heads were also evaluated for 25,365 wells across the country and results are shown in Fig. 13. The results in this case are very similar between the original version and the modified one. Statistics showed a RMSE of 5.5 m in both cases, with the RMSE being dominated by relatively few very large errors while 78 % of the wells have absolute errors

below 5 m. The similarity in simulated groundwater heads between the two model versions indicates that the changes in evapotranspiration patterns have little effect. However, it has to be considered that the simulated groundwater head is controlled by mainly hydraulic conductivity (which does not change between the two versions) and annual recharge upstream of the point of comparison. Since the changes in evapotranspiration patterns mainly effects the summer period, where recharge is low, the effect on annual recharge is limited. In addition, the changes in evapotranspiration patterns will

redistribute recharge patterns, but the combined effect of that at some deeper well filter location will be a mixed signal causing limited changes in groundwater head.

The results of this comparison are promising considering that the model was not re-calibrated with the new inputs. In the future, the model will be recalibrated including a spatial metric as an objective function during the calibration, and it is believed that especially the model bias on discharge can be minimized.

**Conclusions**

In this study the potential of remote sensing outputs to evaluate spatial patterns of hydrological models has been shown. The information derived from remote sensing data can be used as a diagnostic tool for revealing model structural insufficiencies and inconsistencies. Additionally, remote sensing derived variables can be used in hydrological models and hence adding spatial information that is finally translated to the outputs of the models. The use of spatial metrics is beneficial to identify

spatial model deficiencies. Furthermore such metrics are required for a spatial pattern oriented model calibration in order to meaningfully compare the changes in the spatial patterns.

Hydrological model evaluations have traditionally focussed on the temporal dynamics of the outputs and not so much on the spatial component. This study shows that more attention must be given to the spatial pattern evaluation as traditional calibration does not ensure a realistic spatially representation. If the spatial component of the model is neglected, the use of distributed hydrological models is not always meaningful and therefore the use of more simple models could be more

appropriate.

Model re-calibration should focus on a combination of improved parameter regionalization including clay fraction and other derived variables using remote sensing data, spatially countrywide consistency in parametrization and inclusion of dedicated spatial pattern oriented objective functions in combination with discharge and groundwater head observations.

This study was conducted over an energy limited region and over a specific time of the year where ET pays a more important

role in the water cycle. Extending this study to other areas with different ecosystems that combine energy and water limited ecosystems will provide a wider overview on the factors controlling the ET spatial patterns.

**5 Acknowledgments**

The present work has been carried out under the SPACE (SPAtial Calibration and Evaluation in distributed hydrological modelling using satellite remote sensing data) project; The SPACE project is funded by the Villum Foundation

(http://villumfonden.dk/) through their Young Investigator Programme. The authors would like to thank Hector Nieto and Radoslav Guzinski for sharing helpful code in different stages of the study and making the PyTSEB library (https://github.com/hectornieto/pyTSEB). The authors want to thank also to HOBE center for hydrology (http://www.hobecenter.dk/ ) for sharing the eddy covariance data. All MODIS data was downloaded from NASA Land Processes Distributed Active Archive Center (LP DAAC) at the USGS/Earth Resources Observation and Science (EROS)

Center, Sioux Falls, South Dakota, USA.

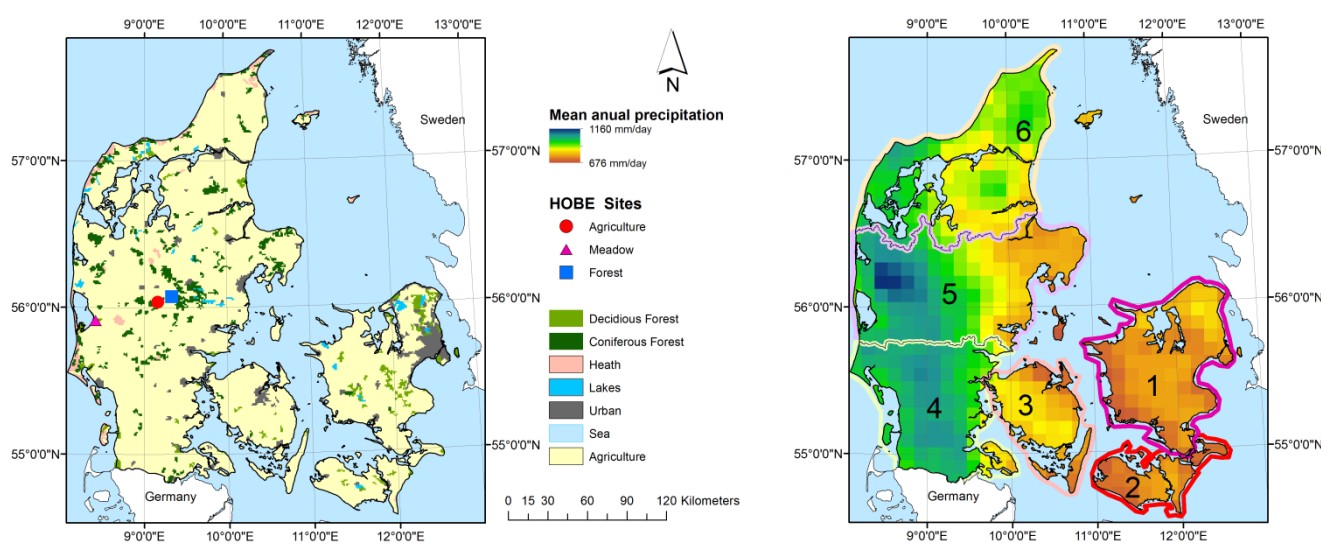

**Figure 1. Presents the study domain. Left map: Land cover map and model domains of the National Water Research Model of Denmark (Bornholm Island excluded in the figure). Right map: National Water Research Model of Denmark domains and mean annual precipitation.**

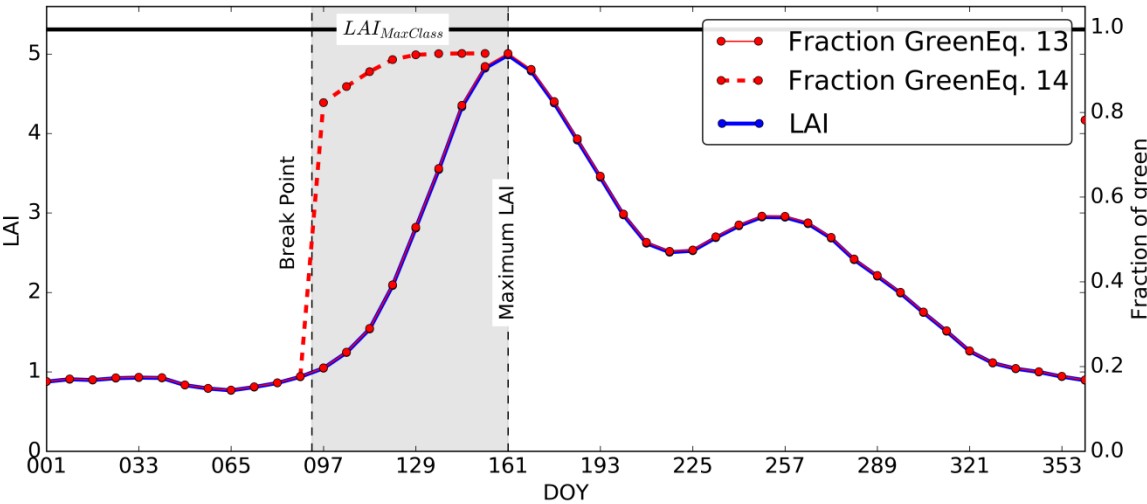

**Figure 2. Diagram of Fraction of Green (Fg) calculation based on the leaf area index (LAI). LAI corresponds to the left ordinate and Fg to the right one. Shadow highlights the region in time were the Eq.13 is replaced by Eq.14. Data presented corresponds to an agricultural pixel from the dataset.**

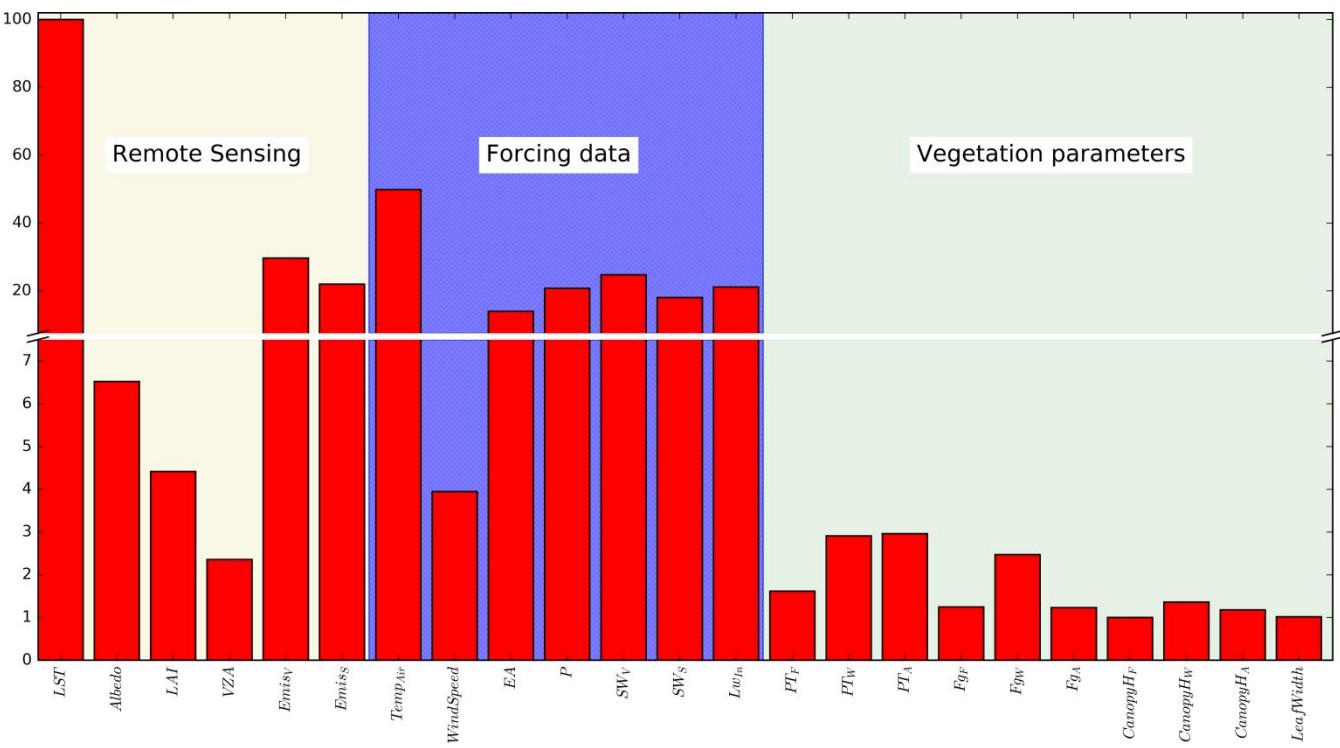

**Figure 3. Sensitivity of 28 TSEB model inputs obtained with PEST. Results are normalized using the most sensitive as reference. Acronims used: LST (Land Surface Temperature), LAI (Leaf Area Index), VZA (View Zenithal Angle), Emiss$_V$ (Emissivity of Vegetation), Emiss$_S$ (Emissivity of Soil), T$_A$ (Air Temperature), EA (Water Vapor pressure above canopy), P (Atmospheric pressure), SW$_V$ (Short wave incoming radiation for vegetation), SW$_S$(Short wave incoming radiation for soil), LW$_{In}$ (Long wave incoming radiation), PT$_F$ (Pristley Taylor parameter for Forest), PT$_W$ (Pristley Taylor parameter for meadow), PT$_A$ (Pristley Taylor parameter for Agriculture), Fg$_F$ (Fraction of green vegetation for forest), Fg$_W$ (Fraction of green vegetation for meador), Fg$_A$ (Fraction of green vegetation for Agriculture), CanopyH$_F$ (Canopy height for forest), CanopyH$_W$ (Canopy height for meadow), CanopyH$_A$ (Canopy height for Agriculture).**

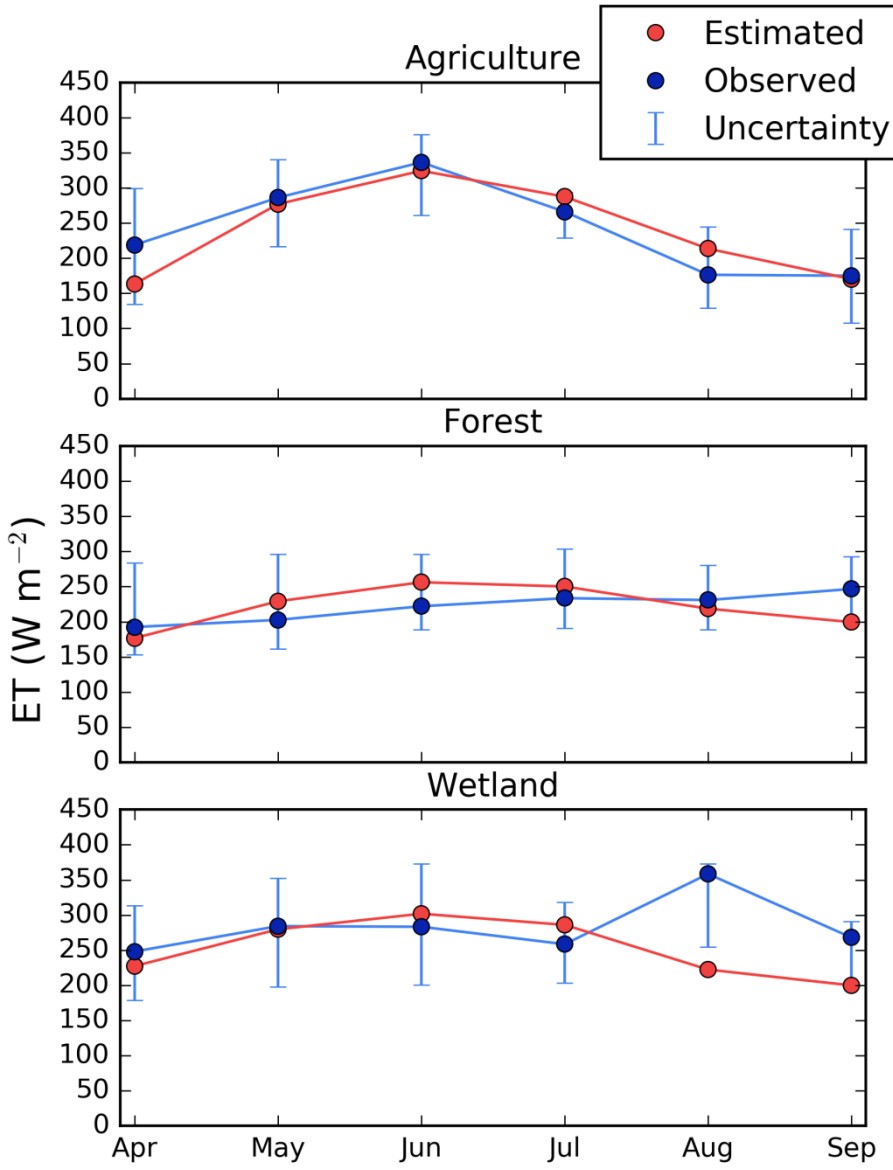

**Figure 4. Comparison of TSEB ET estimates in different land cover types. Uncertainty bars limits represent two situations, the upper in which all the residual is assigned to the latent heat and the lower one in which the residual is assigned to the sensible heat flux.**

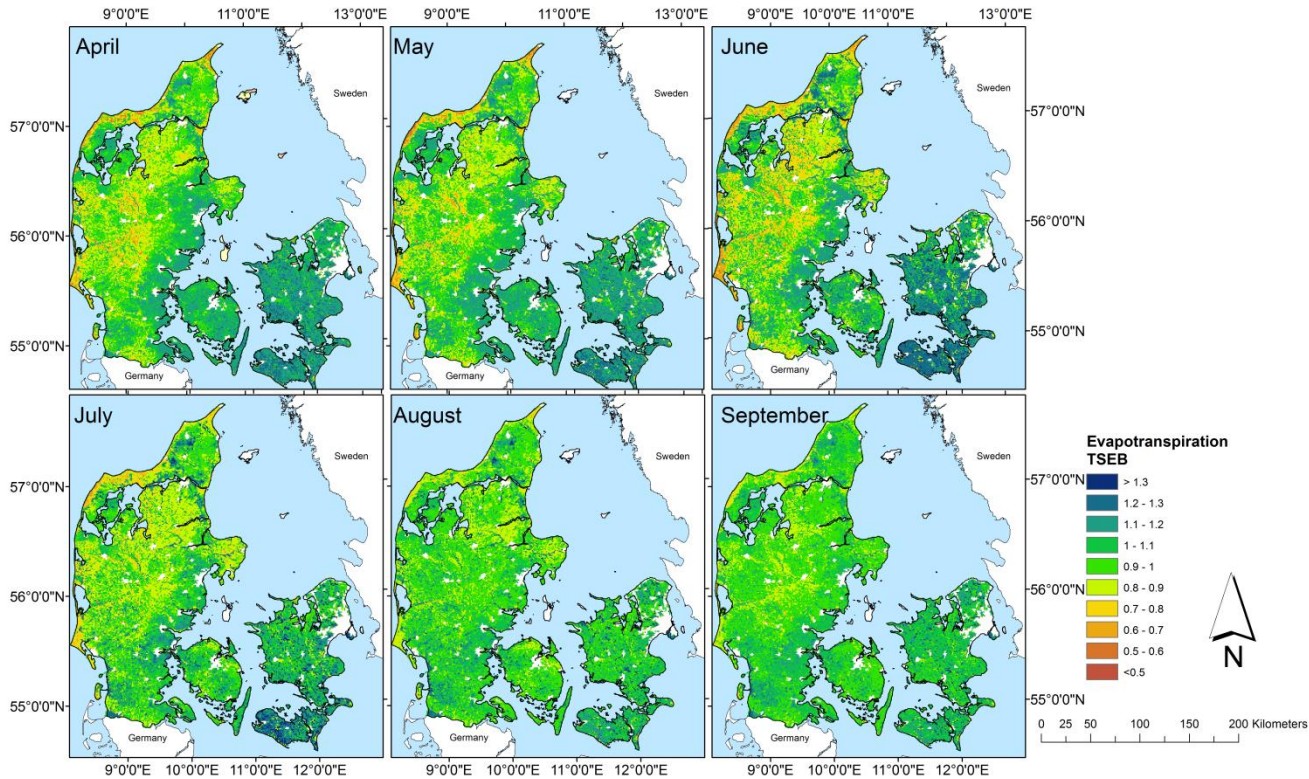

**Figure 5. Mean normalized monthly TSEB ET maps [-] in which urban areas have been masked out.**

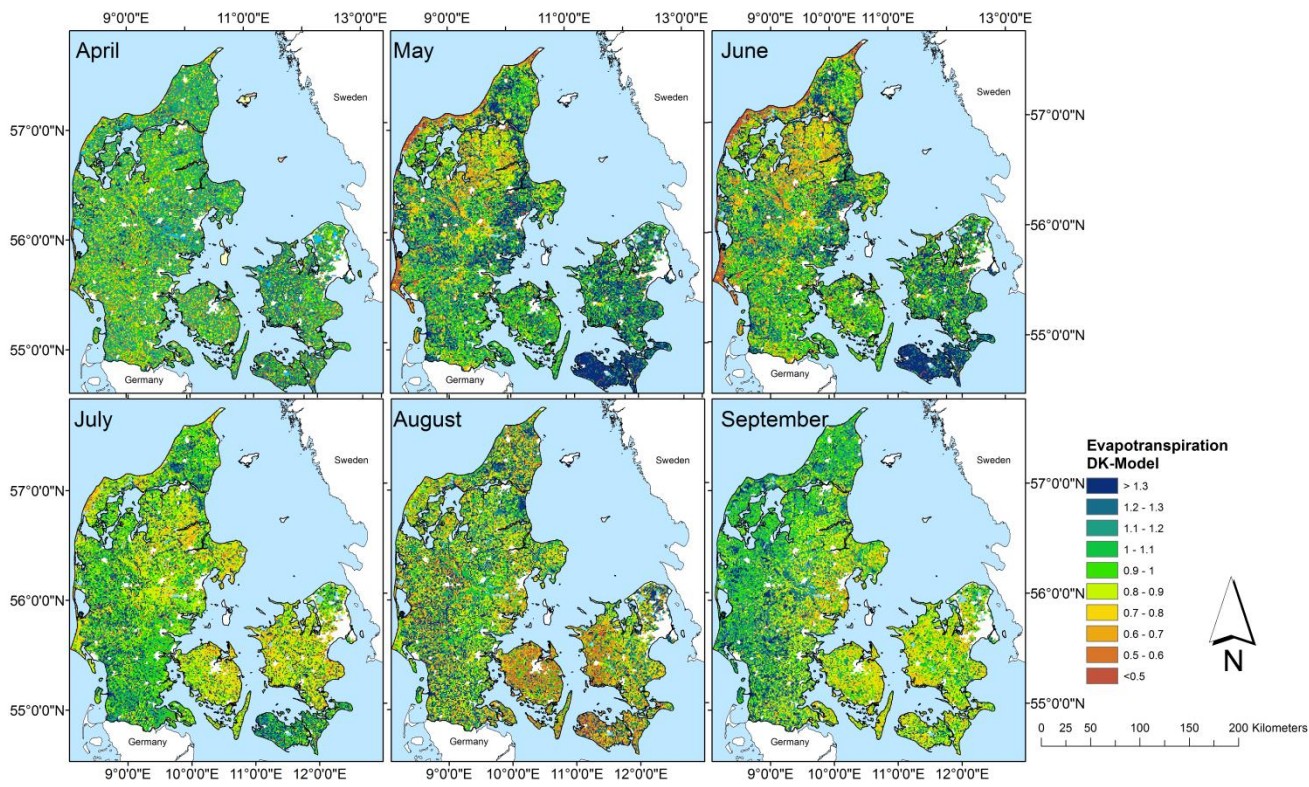

**Figure 6. Mean normalized monthly DK-Model ET maps [-] in which urban areas have been masked out.**

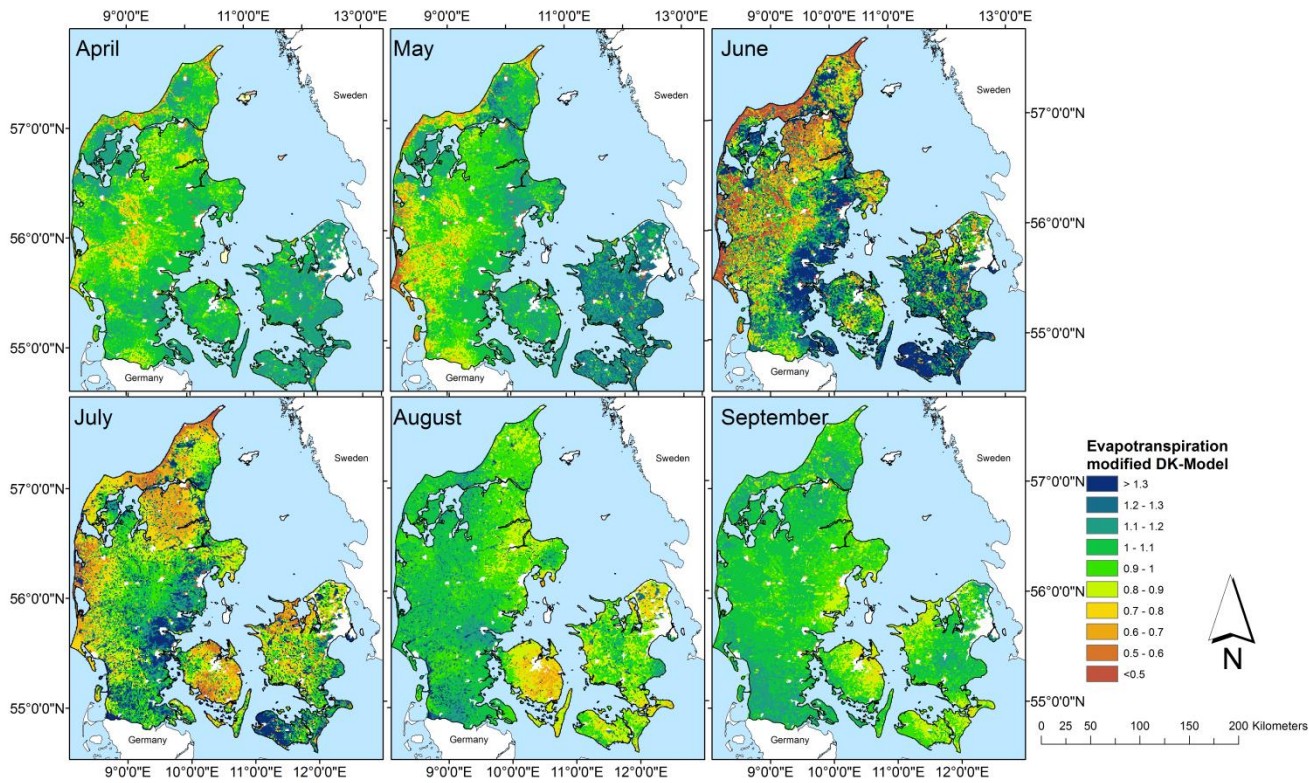

**Figure 7. Mean normalized monthly modified DK-Model ET mean maps [-] in which urban areas have been masked out.**

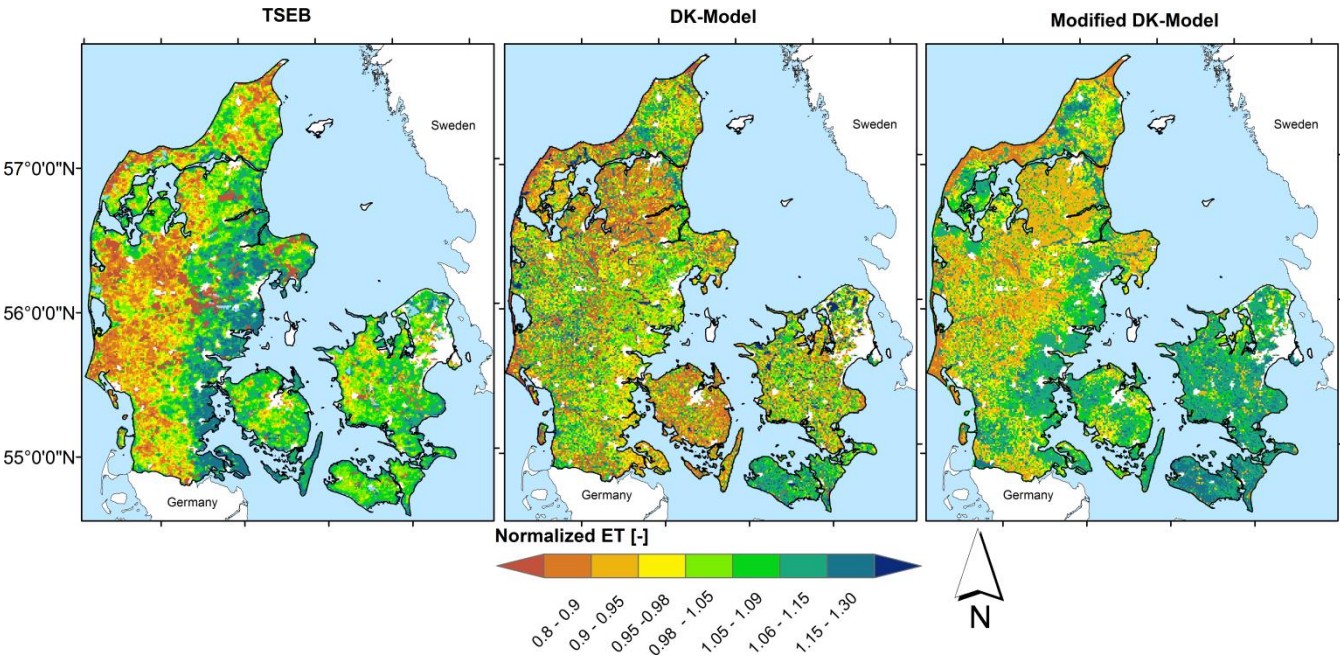

**Figure 8. Normalized growing season maps for the TSEB , DK-Model  and the modified version of the DK-Model. Normalization was conducted dividing the mean map by the mean value of the map.**

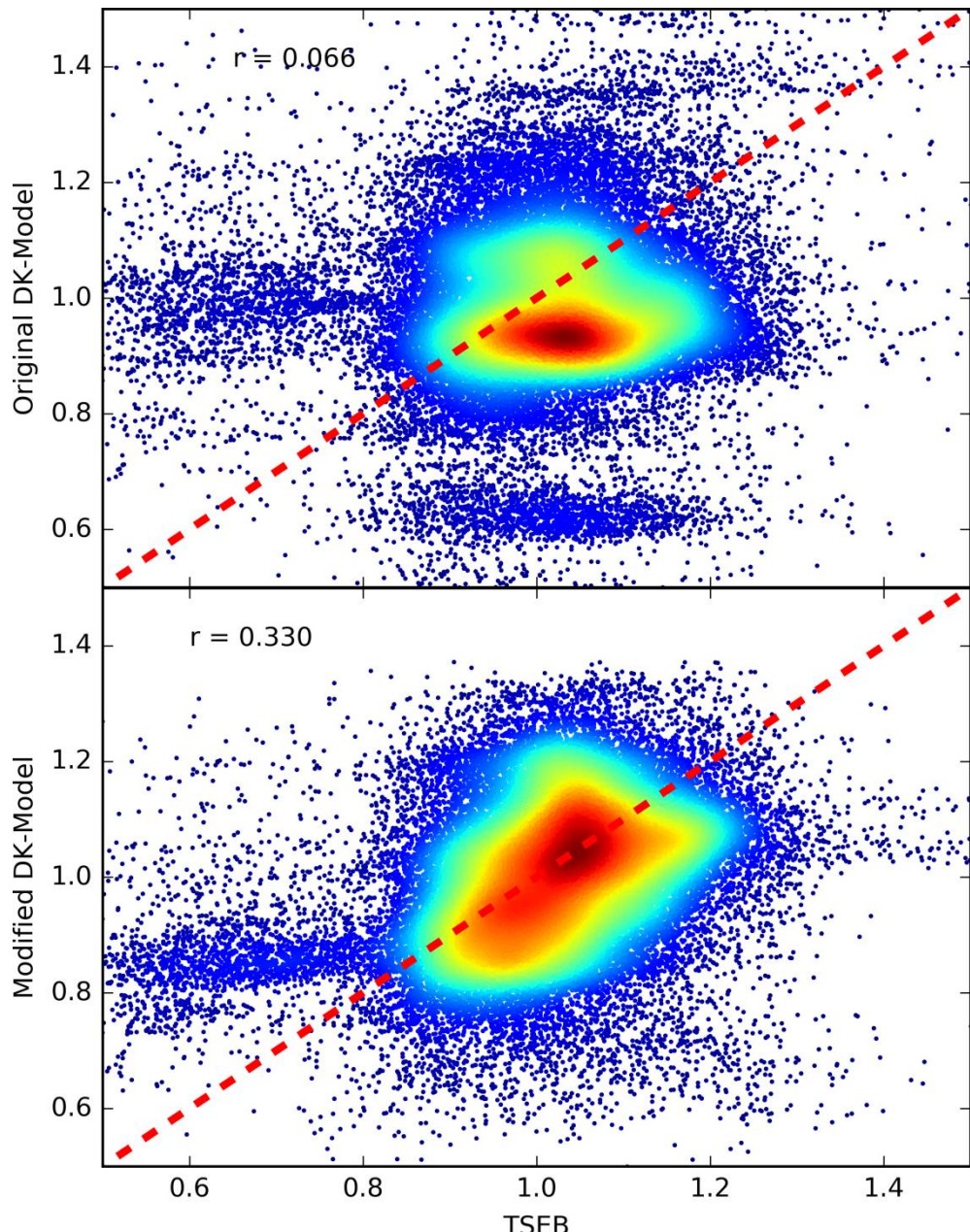

**Figure 9. Scatter plots of normalized ET showing the comparison of the TSEB against the two DK-Model configurations. Upper scatter plot compares the TSEB against the original DK-Model, and lower compares against the modified DK-Model version.**

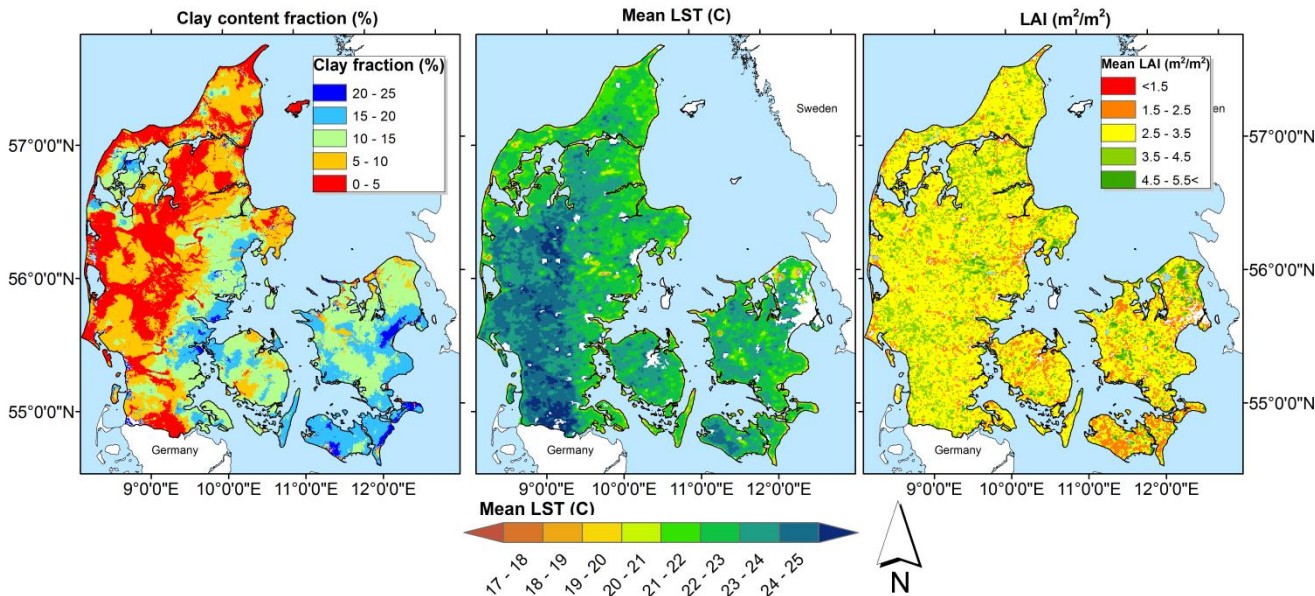

**Figure 10. Maps of three different parameters used in this study. Left map shows the clay fraction distribution (%). Center map displays the mean values of LST (C) during the growing season and right map displays the mean values of LAI (m²/m²) during the growing season and used in the TSEB.**

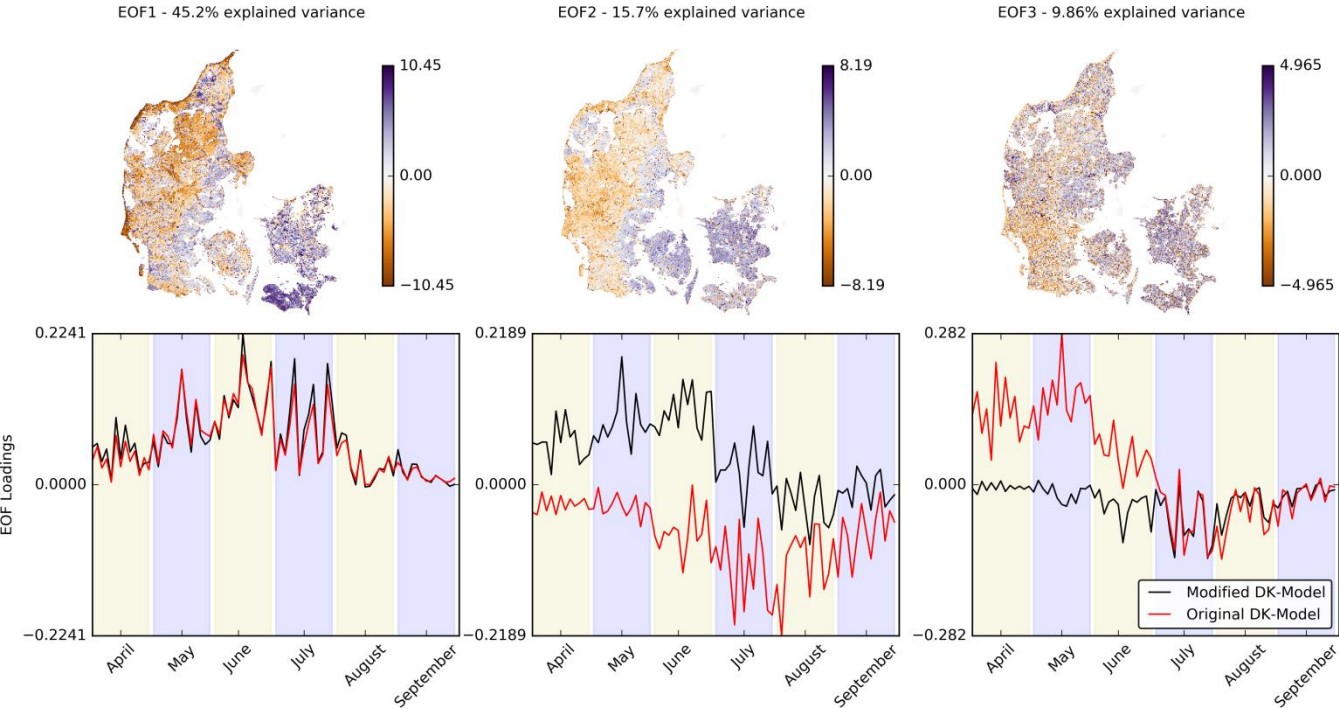

**Figure 11. Maps of the first three EOFs comparing the original setup of the DK-Model and the modified version of it.**

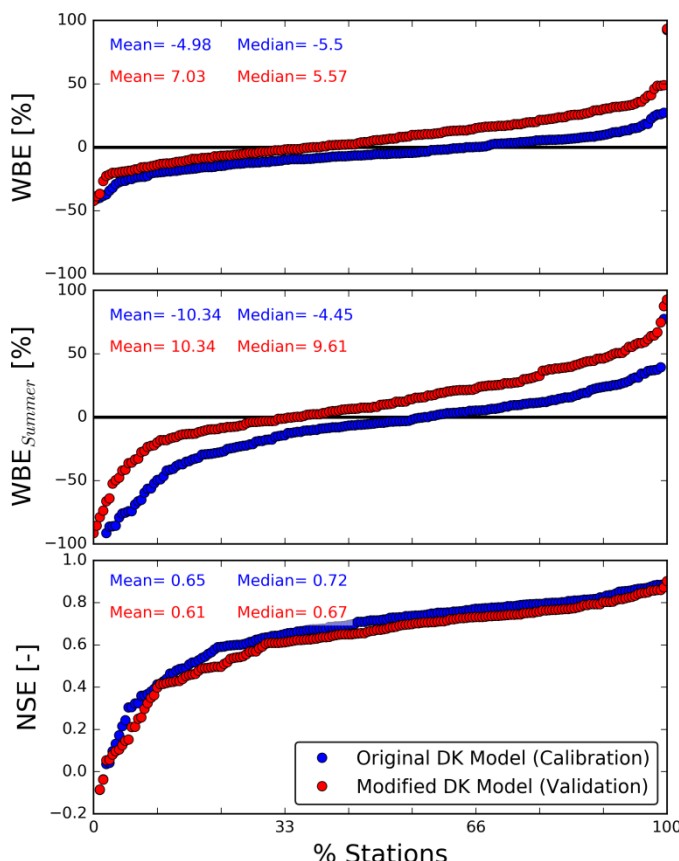

**Figure 12. Model performance statistics showing the results of the original and modified versions of the DK-Model. Stations have been ranked by performance and presented in the horizontal axis as a percentage. Figure shows the Nash-Sutcliffe Efficiency (NSE), water balance error (WBE), and water balance error only for the summer period (WBE$_{Summer}$)**

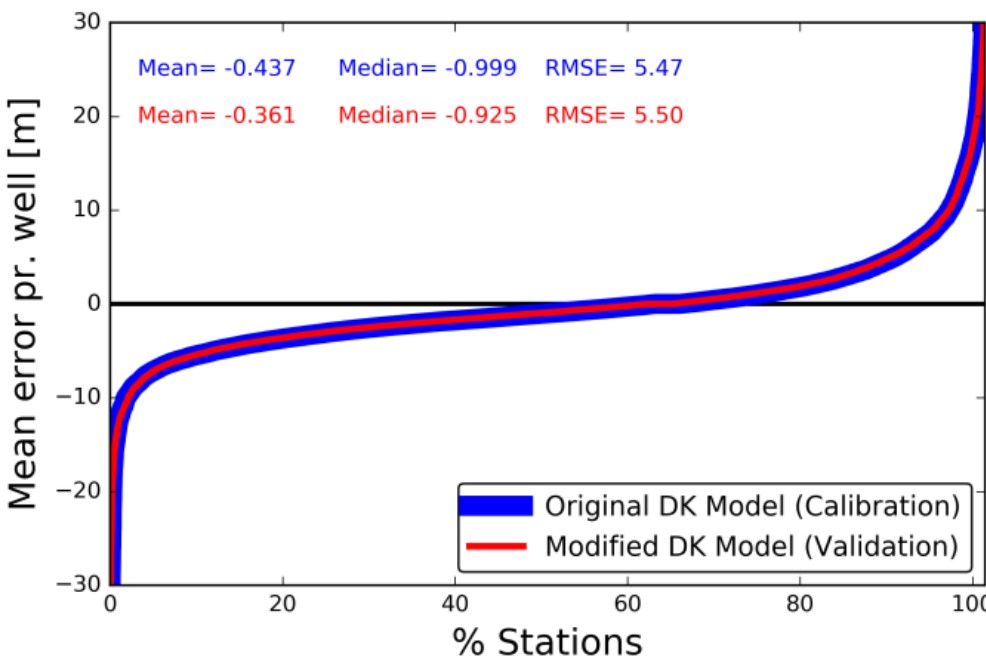

**Figure 13. The figure shows the error and statistics in the ground water heads estimated from the original DK-Model and the modified version of the DK-Model. Stations have been ranked by performance and presented in the horizontal axis as a percentage.**

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
