# Peer review of "Spatial pattern evaluation of a calibrated national hydrological model – a remote sensing based diagnostic approach"

_Hydrology and Earth System Sciences, 2017_

## Referee Comment (RC1) · Anonymous Referee #1 · 5 Jun 2017

**1   General comments**

The manuscript is dealing with the topic of spatial patterns in distributed hydrological modeling. The authors present a study in which they derived a remote-sensing based ET dataset which they use to improve the spatial patterns of the MIKE-SHE national model of Denmark (DK model). These improvements are achieved by adjusting the parameterizations and input data of the existing DK model. They conclude, that spatial patterns of remote sensing data are a valuable information to inform hydrologic models about spatial patterns, whereas these models are usually calibrated on integral signals, such as streamflow.

[Figure]

The topic fits the scope of HESS and is of scientific interest. The chosen methods seems to be appropriate but some clarification in the methods section is still missing. The authors introduce novelty to the field of applying remotely sensed data for hydrological modeling by consideration of bias insensitive pattern matching techniques. The manuscript lacks, here and there, the soundness of the applied evaluations using scientific methods. A lot of evaluation of the spatial patterns in figures 5 to 7 is done on visual basis without proper numerical/scientific quantification. Some features, e.g., the often mentioned "clear" distinction between model region 5 and 6 are hard to observe for the reader if even existing. Further, the adopted DK model is never evaluated regarding streamflow or groundwater levels which is the main purpose/application of this model. Another criticism is the absence of a proper discussion of the findings of this study, there is little referencing to any other study such as Mu et al. (2007, 2011) for remotely sensing based ET estimates.

With exception of section 2.2 the manuscript is well written and good to understand. It could improve by better organization of the sections. I would swap sections 2.3 and 2.4 because sections 2.2 and 2.4 belong together in my opinion. Further, I suggest to fully reorganize and rewrite section 2.2 since it is hard to follow and the storyline is missing in there.

Concluding, I suggest to accept the manuscript for publication in HESS after major revision.

**2 Specific comments**

Introduction:

The introduction is well written and gives a appropriate overview on the topic and shows the novelty of this study compared to existing research.

Methods:

In general section 2.2 should be reorganized and rewritten because it is difficult to follow (I am missing the storyline here) and hard to understand what all the variables and equations are needed for. I think a major thing missing here is the presentation of the TSEB equation to assess which variables are needed in order to estimate ET. This will make clearer why you estimate LAI and vegetation height among others.

- please include TSEB equations

- P4L24: Is the LAI estimate sensitive to its source satellite? So is there any difference in LAI data originating from TERRA compared to AQUA?

- P4L30: What does BRDF mean?

- Eq. 1: Please state the wavelengths for $B_1$ and $B_2$

- clarify for what purpose LAI, albedo, VH and others are needed, I think the TSEB equation will help a lot for that

- Eq. 6: please explain $LAI_{MaxClass}$

- Fig. 2: probably add the growing phase as a gray box, I think the red line should be dotted outside the growing phase as it was estimated with Eq. 5, merge both legends to one, caption: probably show pixel in map (Fig. 1) - row 100 and column 84 definitely means nothing to anybody, add: LAI corresponds to the left ordinate and Fg to the right one.

- Are the data interpolated around the braking point or is a jump appearing in Fg? How reasonable is that?

- P6L5-9: How is that approach justified? Do you have any evidence with observations or references in literature?

- Eq. 8: I think you mean $EF = \frac{ET}{R_n}$

- P6L30-32: I do not understand this sentence

- Eq. 9 and 10: Why was the original RD approach based on LAI adopted to NDVI. LAI is available as seen on previous page. Could you please elaborate a bit on that?

- Eq. 9: Please explain $NDVI_i$ the same as $NDVI_{max}$

- P7L7: LAI in meters?

- How is $RD_{max}$ estimated?

- I don't understand what "matching the original DK model" for RD and KC means. I thought the aim is to make them variable. How did you achieve to make them matching, by parameter calibration? Please elaborate a bit more on that.

- P8L3-6: At P6L14-16: you state an actual value comparison is not anticipated. Here you are calibrating your TSEB model with eddy covariance data. Why? Please elaborate more on that.

- P8L1-2: Wouldn't a variance based sensitivity method better fit the purpose of identifying the parameters which have to be used for model calibration instead of the derivative based approach applied herein? Probably provide some details about the chosen sensitivity approach.

Results and discussion

- P9L5,L8: Please make a distinction between the terms parameter and variable, the reader gets confused otherwise.

- P9L8/Fig 3: better: TempA = $T_a$

- Fig. 3: $LAI_{max}^{Agri}$, $LAI_{max}^{Forest}$, $LAI_{max}^{Meadow}$ do not appear.

- P9L14 & P8L5-6: Why did you select only those 4 parameters out of 10. For PT the others seems to be more sensitive then the forest PT, for example.

- Please justify the assumption to add the residual energy to LE. I only know approaches using corrections based on the Bowen ratio or adding the residual energy to SH.

- Fig. 4: Thanks for including error bars to the plot. I think it is misleading showing only the error bars of the observation. Could you also show error bars on the simulation, e.g., emerging from different parameter sets?

- P9L20: Could you please mentioned the spatial resolutions of EC and RS data?

- The results section is missing in general a discussion with other studies. E.g., estimating ET from MODIS data comparing to Mu et al. (2007, 2011).

- P10L14-15: Is it reasonable to observe lower ET for forest areas? Wouldn't canopy interception increase ET only after precipitation events?

- P10L17: are causing differences in area 2 in the model domain

- P10L19, Fig 6, P11L5, and others: I am very sorry but I cannot observe the pronounced difference between zone 5 and 6. Could you provide some more information on that, e.g., zoomed plot numerical analysis? At the provided plots I do not see this features.

- P10L10: reformulate: extracted

- P10L24: ... does not necessarily lead to reasonable ET ...

- Fig. 9 and P11L9-15: Could you provide numerical evidence to the explanatory variables of the spatial patterns of ET. I can see the E-W gradient in clay content and ET but the others are not observable. Consider rewriting or deleting some of your conclusions since they are not supported by your data. Possibilities to get evidence: scatterplots or SPEARMAN rank correlations.

- I miss the comparison of the model performance in streamflow and groundwater table between the original and modified DK model. I understand that the spatial representativeness of the modified DK has improved compared to the original one. But shouldn't be made sure that the water balance is still sufficiently represented by assessing the streamflow and groundwater tables since that is the major purpose of the model? Therefore, the model performance shouldn't deteriorate significantly if evaluated with those variables.

**3 Technical corrections**

- Fig. 1: excluded in figure) - parentheses missing, consider using different symbols for Agri and Meadow because they are hard to distinguish.

- P2L10: rational behind developing

- P2L21-25: because you do not provide exhaustive list of references for each application example i suggest to use 'e.g.,' in front of the references

- P3L7: Figure 1 presents the herein used study domain.

- P4L27: I would put the LAI sentence to the previous paragraph and start the new paragraph with: "The study focuses"

- P5L6: delete successfully after Boegh et al.

- P5L7: this study instead of the study - you should check that in the entire manuscript

- P5L7: similar approach was applied where ... - please delete "was applied" later in the sentence

- P5LL25:please do not introduce abbreviation like 10U which are never used in the manuscript

- P6L5: To identify the different periods, first, the dates ...

- Is $LAI_{i.max}$ the same as $LAI_{Max}$ in Eq. 4 and Eq. 9? check consistency

- P6L6: breakpoint Fig. 2 not figure 3

- P6L6: better: breakpoint Fig. 2, i.e., the onset of the growing season

- P6L8: Eq. 6 instead of 5

- you are switching from Eq. to equation and Fig. to figure in the entire manuscript check consistency

- probably check for figure and equation referencing in the entire manuscript

- Eq. 7 & 8: netRad and net radiation - consistency

- I would suggest to use formula symbols like $R_n$ instead of words like netRad

- P6L20: The resulting maps ...

- P6L21: in just climatological maps

- P6L26: latent heat (LE) or **evapotranspiration** measurements **are**

- P6L29: which is usual instead of not unusual

- P7L7: RDi = RD$_i$

- P10L11: Fig. 5 instead of Fig. 56

- P10L11: pattern identifie**d** the **T**SEB

- P12L22: the meso.. instead of The meso..

---

## Referee Comment (RC2) · Anonymous Referee #2 · 22 Jun 2017

Major comments:

Mendiguren et al. studied the importance for a hydrological model simulation to reproduce similar spatial patterns as those of remote sensing data. Their modified version of the DK-model provides a simulated evaporation result that has more similar spatial features found in the remote sensing based ET. Generally, I read the paper with great interest. The paper fits very well within the stated scope of journal. I consider that the evaluation of spatial patterns of model simulation result is still quite novel and rarely done in common hydrological model practices. However, there are still some major issues to be addressed before this manuscript is being accepted for publication.

[Figure]

The authors present an improved model in which remote sensing derived data were used for parameterizing vegetation parameters. They claim that the improved model provides better results as it has more similar spatial patterns as those of remote sensing data. However, it seems that the benchmarking of simulation results is limited to the evaporation flux only (especially its spatial pattern). I suggest performing more evaluation and comparison to the 'original' and 'improved' model simulation results, particularly to their discharge and groundwater head results.

In addition, I fell that the presentation, writing and structure of the paper must be upgraded. Often, there is no clear gap/interval between paragraphs. There are paragraphs and sentences do not flow with their previous ones. These make the paper difficult to read and understand in quite a lot of places. Moreover, in the Introduction section, I hardly find any sentences related to the actual or the main objective of the study (which is to evaluate spatial patterns of a model simulation result?). I also think that the presentation and structuring of the Methods section must be improved. Furthermore, I recommend to have separated sections of Results and Discussion. Please also see some suggestions in the following minor comments.

Minor comments:

Page 1, lines 13-15: "The main hypothesis of the study is . . . ." I suggest rephrasing this sentence. Moreover, I could not find this hypothesis in the Introduction section.

Page 1, lines 26-28: Using your modified version of the DK-model, did you get any improvements on your discharge and groundwater head simulation results?

Page 2, lines 21-24: I suggest including some references about the applications of using satellite data for assessing groundwater as well (e.g. Rodell et al., 2009, http://dx.doi.org/10.1038/nature08238; Sutanudjaja et al., 2013, https://doi.org/10.1016/j.rse.2013.07.022; Richey et al., 2015, http://dx.doi.org/10.1002/2015WR017349). I guess that they are relevant for your study as you use the DK-model that simulates groundwater head dynamics. I am also

curious how the improvement that you introduced (based on remote sensing data) affects groundwater head simulation.

Page 3, lines 1-3: Could you please elaborate more with what you meant by the "eminent risk" here? Some references will be helpful.

Page 3, lines 14-16: Could you please elaborate more with what you meant by the "diagnostic approach"?

Page 3, lines 26-27: Neglecting biases/differences in the absolute values is a quite brave assumption. Could you please provide some justification behind it? Why?

Page 3, line 30: This should be "Sections 2.2 and 2.3".

Section 3: Please consider to reorganize the structure (sub-sections) of Section 2. I found that it is quite difficult to understand and follow the sequence of each step of your methods.

Page 4, lines 10-11: You have discharge and groundwater head measurement data. Could you please evaluate the results of your improved model to these data?

Page 4, line 19: "TSEB can successfully be applied with a single LST observation, ... ." This sentence is not clear for me. What do you mean by "single LST observation"?

Equation 1: Please put the reference to this equation. Moreover, it will be very helpful if you include the unit or the dimension for every variable. Example: NDVI [-]; LAI [-]; VH [unit: m]

Page 5, line 1: "... MODIS band number." Please be more specific with these "band numbers".

Equations 2 and 3: Please put the reference to these equations (e.g. as you introduced the reference for Equation 4).

Page 5, line 10: Please include the dimension/unit for the parameters alpha and beta.

I guess they are dimensionless.

Page 5, line 18: This sentence does not flow well with the ones before it.

Page 5, line 21: Could you please elaborate more with what you meant by the "highest quality pixels"?

Page 5, line 27: I miss the explanation why you have to calculate "Fraction of Green"? For what purpose?

Page 5, line 27 to Page 6, line 9: Please rewrite this part. I hardly follow it. And can you please justify this assumption to the reality?

Page 6, line 10: This sentence does not flow well with the ones before it.

Equations 7 and 8: What is the difference between "Net Radiation" and "netRad". If they are the same, please be consistent with your variable names. Please also include the dimension/unit.

Page 6, lines 22-23: I hardly understand this sentence.

Page 6, lines 24-32: "Data from three eddy covariance (EC) flux towers is used as a reference to perform a sensitivity analysis and calibration of some of the vegetation parameters of the TSEB." I guess that the methods for sensitivity analysis and calibration are given in the Section 2.4? Please consider to reorganize Section 2 so that all your method steps are presented in a logical sequence.

Section 2.3: It makes more sense to put this Section 2.3 directly after Section 2.1.

Page 7, lines 8-9, Equation 9: Why did you have to substitute LAI with NDVI?

Equations 11 and 12: Please put some references.

Page 8, line 1: ". . . perturbation with respect to a change in model performance." What is your model performance objective function? RMSE? NSE? KGE?

Page 8, line 5: How did you choose these four parameters?

Equation 13: What is the best value of SEOF? 0? Please clarify.

Section 3: I recommend to have separated sections for Results and Discussion.

Section 3.1: Sensitivity analysis: Can you please explain more about how you perturb your model input data? What is their range for the maximum and minimum values of each variable?

Page 8, line 14: How did you choose these four parameters? Based on your sensitivity analysis (Fig. 3)? How?

Section 3.1: TSEB calibration: How realistic is your calibrated TSEB result map (including its spatial pattern)? Did you compare it to other studies (e.g. to MODIS, GLEAM, etc.)?

Figures 5, 6 and 7: Please use the same and consistent legend (color and values) for all figures so that they can easily be compared.

Page 9, lines 25-28: "The main aim of TSEB . . . ." It seems that the sentences do not flow with their previous ones.

Figure 8: How did you normalize all three maps? What was the motivation to normalize these three maps?

Figure 10: Please improve the caption for Fig. 10. What does the color mean here?

Page 10, line 16: MIKE-SHE = DK-model?

Page 11, lines 1-15: Please rephrase the sentences in these lines. This seems a very important finding in your result. I guess that this result appears very dependent on your choice to use Equation 11 (Page 7, line 19). I am wondering how you derive this equation, particularly their factor 12 and constant 0.2? Can we implement this equation for other study areas, e.g. to other climate regions (e.g. tropical areas).

Page 12, line 27 to Page 13, line 16: I suggest starting a new subsection for this part.

Others:

Figure 1: For the figure on the right, what do the numbers (1 to 6) stand for?

Figure 2: Please indicate the pixel (row 100, column 84) in Fig. 1.

———————————————

---

## Author Comment (AC2) · 6 Jul 2017

Correspondence to Gorka Mendiguren (gmg@geus.dk)

Response to anonymous Referee #1

**Reply:** The authors would like to thank the reviewer for his/her detailed and elaborated review of the manuscript. The comments and suggestions are very much taken into thorough consideration as we believe they will improve the reading and add a significant contribution to the manuscript increasing the scientific quality. We are very pleased to read that he/she considers the manuscript appropriate for publication after major revision in Hydrology and Earth System Sciences (HESS). We hope that the changes conducted in the revised version of the manuscript will be well received by the reviewer and that he/she will regard the publication as fit for submission in Hydrology and Earth System Sciences.

**1 General comments**

The manuscript is dealing with the topic of spatial patterns in distributed hydrological modeling. The authors present a study in which they derived a remote-sensing based ET dataset which they use to improve the spatial patterns of the MIKE-SHE national model of Denmark (DK model). These improvements are achieved by adjusting the parameterizations and input data of the existing DK model. They conclude, that spatial patterns of remote sensing data are a valuable information to inform hydrologic models about spatial patterns, whereas these models are usually calibrated on integral signals, such as streamflow.

The topic fits the scope of HESS and is of scientific interest. The chosen methods seems to be appropriate but some clarification in the methods section is still missing. The authors introduce novelty to the field of applying remotely sensed data for hydrological modeling by consideration of bias insensitive pattern matching techniques. The manuscript lacks, here and there, the soundness of the applied evaluations using scientific methods. A lot of evaluation of the spatial patterns in figures 5 to 7 is done on visual basis without proper numerical/scientific quantification. Some features, e.g., the often mentioned "clear" distinction between model region 5 and 6 are hard to observe for the reader if even existing. Further, the adopted DK model is never evaluated regarding streamflow or groundwater levels which is the main purpose/application of this model. Another criticism is the absence of a proper discussion of the findings of this study, there is little referencing to any other study such as Mu et al. (2007, 2011) for remotely sensing based ET estimates. With exception of section 2.2 the manuscript is well written and good to understand. It could improve by better organization of the sections. I would swap sections 2.3 and 2.4 because sections 2.2 and 2.4 belong together in my opinion. Further, I suggest to fully reorganize and rewrite section 2.2 since it is hard to follow and the storyline is missing in there. Concluding, I suggest to accept the manuscript for publication in HESS after major revision.

**Reply:**

We agree with the reviewer that the inclusion of statistics of model performance regarding streamflow and groundwater head will add significantly to the manuscript. Initially we decided that the comparison between the performance of the original and modified DK-model would not be fair, since the original model has been calibrated against streamflow and head, whereas the modified has not (single model run time of around 40 hours makes re-calibration a huge task). However we have decided to include a full section comparing the streamflow statistics, water balance errors and groundwater head errors with the original and modified version of the DK model in the manuscript.

Regarding the numerical quantification of the spatial patterns similarities, this is something our group has worked extensively on (Koch and Stisen, 2017) and we currently have another manuscript under review that addresses this issue. However, for this particular study the detailed numerical quantification of the pattern similarities were believed to be a bit irrelevant because the differences are so large between the TSEB and DK-model and therefore a visual interpretation was thought as more appropriate. However we did quantify the correlation coefficients in Figure 10. In a revision we will also add correlations between TSEB AET and different variables (fig 9) to provide a more numerical quantification.

Regarding the description of the differences between model domains, we agree that this has not been written very clearly and this will be improved during revision. But the point was that the lack of spatial consistency in model parametrization results in simulated pattern artifacts. This is still the case and will be exemplified better.

Regarding Mu et al., YEAR we are aware of that product, but chose to produce a satellite based AET dataset based on land surface temperature (LST), which is not included in the Mu model. Oppesed their model is primarily driven by LAI and NDVI. This choice is justified by the fact that in our study the LST/TSEB and NDVI/LAI patterns are sometimes very different, especially during the senescence phase. We will add an elaborated description of the TSEB-method and discuss how it differs from other regional AET methods and argue why we chose the TSEB.

Koch, J., and Stisen, S.: Citizen science: A new perspective to advance spatial pattern evaluation in hydrology, PLOS ONE, 12, e0178165, 10.1371/journal.pone.0178165, 2017.

**2 Specific comments**

Introduction: The introduction is well written and gives a appropriate overview on the topic and shows the novelty of this study compared to existing research.

Methods: In general section 2.2 should be reorganized and rewritten because it is difficult to follow (I am missing the storyline here) and hard to understand what all the variables and equations are needed for. I think a major thing missing here is the presentation of the TSEB equation to assess which variables are needed in order to estimate ET. This will make clearer why you estimate LAI and vegetation height among others.

- please include TSEB equations

**Reply:** We agree that the method section should be reorganized, and this will be done by clearly splitting the TSEB model and the hydrological model in two sections with the appropriate subsections on methodology, remote sensing derived inputs and calibration/validation setup.

The TSEB method cannot be summarized into a single equation, but we understand that it becomes difficult for the reader to follow the TSEB model without some extra explanation. Therefore following the reviewer recommendation we have extended the model description, included several equations that hopefully helps the reader to understand how TSEB works.

However, the model implies several steps and some theory is involved as well so the reader is encouraged to read the article from Norman et al. 1995 to get a complete idea of how the model works in detail.

The text will be modified and the description of the model gets in the manuscript will get this:

**"2.2 TSEB setup and remotely sensed derived inputs**

*The Two Source Energy Balance Model (TSEB) proposed by Norman et al. (1995) is used to retrieve mean monthly maps of ET across Denmark. In our study we have incorporated the code which is provided by the pyTSEB package (https://github.com/hectornieto/pyTSEB last accessed 30/01/2017). The applied model is a two layer model that treats soil and vegetation separately and estimates fluxes on the basis of LST and air temperature ($T_{Air}$) among other input variables. As presented in Norman et al. (1995), and presented here in a very simplified explanation. The model is based on the energy balance equation:*

$$LE = R_n - H - G$$

*Where H is the sensible heat flux, G is the ground heat flux, $R_n$ is the net radiation and LE is the latent heat flux.*

*The sensible heat flux (H) is calculated  as:*

$$H = H_C + H_S = \rho C_P \left[ \frac{T_C - T_A}{R_A} + \frac{T_S - T_A}{R_A + R_S} \right]$$

*Where $H_C$ and $H_S$ are the sensible heat flux for the canopy and soil respectively,  $T_C$ and $T_S$ are the canopy and soil temperatures, $\rho C_P$ is the volumetric heat capacity of air,  $R_S$ is the resistance to heat flow in the boundary layer above the soil surface and $R_A$ is the aerodynamic resistance expressed as:*

$$R_A = \frac{\left[ ln\left(\frac{z_U - d}{z_M}\right) - \Psi_M \right]\left[ ln\left(\frac{z_T - d}{z_M}\right) - \Psi_H \right]}{0.16U}$$

*where $z_U$ and $z_T$ are the height of the wind speed and air temperature (U and Ta)is the aerodynamic resistance, $R_S$ is the resistance to heat flow in the boundary layer immidiatelly above the soil layer. $T_A$ represents the air temperature, $T_C$ is the canopy temperature, $T_S$ is the soil temperature. $\Psi_M$ and $\Psi_H$ are the diabatic correction factors for momentum and heat, d is the displacement height ($d \approx 0.65\ h_c$, and $h_c$ is the height of the canopy), $z_M$ is the displacement height for momentum ($z_M \approx\ h_c/8$).*

*The model starts with an iterative process in which it finds the $T_C$ which satisfies the energy balance equation. The divergence of net radiation in the canopy ($\Delta R_n$) is used to partition sensible and latent heat fluxes using the Priestley-Taylor approximation (Priestley and Taylor, 1972) for the green part of the canopy. The transpiration is given by the next eq.*

$$LE_C = 1.3f_g \frac{S}{S+\gamma} \Delta R_n$$

*Where $f_g$ represents the fraction of LAI that is green, S is the slope of the saturation vapor versus temperature curve and $\gamma$ is the psychrometric constant and where $\Delta R_n$ is calculated as:*

$$\Delta R_n = R_n - R_n exp(0.9 ln(1-f_c))$$

*And where $f_c$ is calculated as:*

$$f_c = 1 - exp\left(\frac{-0.5 LAI}{cos\,\theta}\right)$$

*Where $\theta$ is the viewing angle.*

*The model iterates until the energy balance equations are satisfied for soil and canopy. The readers referred to Norman et al. (1995) to find a fully description of the model for a more detailed explanation of how the iteration process is carried out.*

*Several inputs to TSEB are directly obtained from the LST product from the Moderate Resolution Imaging Spectroradiometer (MODIS) sensor all at 1 km spatial resolution; day time LST and day time VZA obtained from MOD11A1 and MYD11A1 products flown on TERRA and AQUA respectively. The decision of whether to use LST from TERRA or AQUA is based on the percentage of high quality pixels available covering Denmark in each scene. The quality flags included in the products is used to select only those pixels with the best observation possible. LAI is derived using an empirical relationship with the Normalized Difference Vegetation Index (NDVI) (Rouse et al., 1973)."*

Norman, J. M., Kustas, W. P., and Humes, K. S.: Source approach for estimating soil and vegetation energy fluxes in observations of directional radiometric surface temperature, Agricultural and Forest Meteorology, 77, 263-293, http://dx.doi.org/10.1016/0168-1923(95)02265-Y, 1995.

Priestley, C. H. B., and Taylor, R. J.: On the Assessment of Surface Heat Flux and Evaporation Using Large-Scale Parameters, Monthly Weather Review, 100, 81-92, 10.1175/1520-0493(1972)100<0081:OTAOSH>2.3.CO;2, 1972.

Rouse, J. W., Haas, R. H., Deering, D. W., and Schell, J. A.: Monitoring the vernal advancement and retrogradation (green wave effect) of natural vegetation, Goddard Space Flight Center, Greenbelt, MD, 87, 1973.

• P4L24: Is the LAI estimate sensitive to its source satellite? So is there any difference in LAI data originating from TERRA compared to AQUA?

**Reply:** I think you mean the LST instead of LAI (P4L24). TERRA and AQUA satellites have identical characteristics regarding spectral wavelengths of the sensor they carry. Both sensors operate together with different overpass times and therefore the difference in LST is mostly due to the overpass time, not due to sensor.

Regarding the estimation of LAI the MODIS product that we used combines information from both satellites, TERRA and AQUA.

• P4L30: What does BRDF mean?

**Reply:** we apologize for the typo. We will included in the text what BRDF means:  Bidirectional Reflectance Distribution Function

• Eq. 1: Please state the wavelengths for B1 and B2

**Reply:** Wavelengths have been included in the text. B1=645.5nm and B2= 856.5 nm.

• clarify for what purpose LAI, albedo, VH and others are needed, I think the TSEB equation will help a lot for that

**Reply:** We will include a more detailed explanation of the TSEB that hopefully helps to clarify this question. We will also add a brief description in the text indicating how the albedo is used to calculate the net radiation.

Albedo was used in the study to calculate the net radiation.

• Eq. 6: please explain LAIMaxClass

**Reply:** LAIMaxClass, is the maximum LAI value for a given land use class. This value is used to scale any given LAI value for a particular grid and day relatively to the land use class it belongs to**).** We will include the explanation of the LAI MaxClass in the manuscript.

• Fig. 2: probably add the growing phase as a gray box, I think the red line should be dotted outside the growing phase as it was estimated with Eq. 5, merge both legends to one, caption: probably show pixel in map (Fig. 1) - row 100 and column 84 definitely means nothing to anybody, add: LAI corresponds to the left ordinate and Fg to the right one.

**Reply:** We have followed the reviewer recommendation and added a grey box in the figure. The figure looks now like this.

[Figure]

Caption in the text has been modified following the suggestions and now looks like this:

*"Figure 2. Diagram of Fraction of Green (Fg) calculation based on the leaf area index (LAI). LAI corresponds to the left ordinate and Fg to the right one. Data presented corresponds to an Agricultural pixel from the dataset. Grey area shows the region where Eq. 6 is used instead of Eq.5*

• Are the data interpolated around the breaking point or is a jump appearing in Fg? How reasonable is that?

**Reply:** The data is not interpolated in the breaking point. By the approach we implemented in the study, we aimed at detecting the beginning of the growing season by finding the point in which the LAI presented an increased by 20% compared to the pre-growing season low. Once the date where that happened was found we assumed that all the fraction of the vegetation should present most of the fraction green and therefore we considered that a rapid change reaching values close to 0.9 were meaningful. Then the Fg values follow the evolution of the LAI and decreases as soon as the senescence of the vegetation starts and LAI decreases.

• P6L5-9: How is that approach justified? Do you have any evidence with observations or references in literature?

**Reply:** The approach is not found in the literature, we will explain better why we apply it. We agree with the reviewer that the explanation of what we conducted to retrieve the Fg was very difficult to understand and read (in addition we also had put in a wrong equation). We have completely rewritten this section hoping to make it clearer for the reader. The new text says:

*"....Fraction of Green vegetation was derived from LAI following the next equation:*

$$Fg_i = \frac{LAI_i}{LAI_{MaxClass}} \text{ (EQ 5)}$$

*Where Fgi indicates the Fraction of green for a certain pixel i, LAIi indicates the LAI value for a pixel i  and $LAI_{MaxClass}$ is the maximum LAI value for an specific land cover type. This equation was applied to needle leaf forest land cover type.*

*For the other land cover types (deciduous, grasslands, crops…) equation xxx was modified adding another term. These land cover types show a stronger seasonality. In order to represent the strong difference in fraction of green vegetation between the period before and after senescence we introduced a different equation for the period between crop emergence and senescence, where we assigned higher values of $F_g$ to non-needle leaf forest land covers, Figure xx.   For these types of vegetation Fg will be allowed to increase rapidly just after crop emergence by substituting EQ 5 by EQ 6.*

$$Fg_i = \frac{LAI_{i,max}}{LAI_{MaxClass}} \cdot (1 - e^{(-2 \cdot LAI_i)}) \qquad (EQ\ 6)$$

*Where $LAI_{i,max}$ indicates the Maximum LAI value for a pixel i.*

*This substitution is only conducted during part of the phenological year, more specifically for the period defined by an increase in 20% increase in LAI compared to the winter low and until the to the time at which LAI reaches its maximum (see next Figure )."*

[Figure]

We believe these assumptions fits well with reality, since a given LAI value before and after senescence can have quite different Fg values. During the growing season most of the plant remains green, which is quite well represented with the modification including the exponential term in the equation. After the point, where vegetation has reached its maximum seasonal development (we assume at  maximum LAI is maximum), senescence starts and more non- photosynthetically active regions start to appear in the plant, what is translated in lower Fg values.

• Eq. 8: I think you mean EF = ET/Rn

**Reply:** Yes we meant that, we apologize for the mistake. It has been also changed in the text.

• P6L30-32: I do not understand this sentence

**Reply:** We have modified the sentence as it was confusing. In the manuscript it was written as:

*"The observed values used during TSEB evaluation are the Bowen ratio (Bowen, 1926) corrected values and the associated uncertainty estimate is the span between all error in the closure problem being assigned to the latent heat and no error in the latent heat."*

And now is written as:

*"The evaluation of the TSEB was conducted using as reference the data of the EC systems from the 3 different land cover types that were corrected for the energy closure using the Bowen ratio (Bowen, 1926). The uncertainty of the energy closure issue spam between all closure errors being assigned to sensible heat for the first limit and for the second limit all error being assigned to latent heat."*

Bowen, I. S.: The ratio of heat losses by conduction and by evaporation from any water surface, Physical Review, 27, 779-787, 10.1103/PhysRev.27.779, 1926.

• Eq. 9 and 10: Why was the original RD approach based on LAI adopted to NDVI. LAI is available as seen on previous page. Could you please elaborate a bit on that?

**Reply:** In the first stages of the study we used the approach based on LAI but we found some problems when utilizing that equation to translate LAI to root depth. In this study the LAI was developed on an empirical relationship based on an exponential equation on NDVI and LAI. When converting LAI to root depth that resulted in very abrupt temporal transitions in root depths during the year. Therefore we change the root depth calculation to be a function of NDVI in order to keep the simple linear formulation in eq. 9/10.

• Eq. 9: Please explain NDVIi the same as NDVImax

**Reply:** I think you mean eq 10**.** Sorry for not including it. Now it is included in the manuscript.

Now the text is as follows:

*"Where $NDVI_i$ is the value of the NDVI for a pixel i, and $NDVI_{max}$ indicates the maximum NDVI value for the pixel in the time series...."*

We have double checked this error in the manuscript and correct it where it has been necessary.

• P7L7: LAI in meters?

**Reply:** Sorry, only root depth in meters, LAI in [m2/m2]

• How is RDmax estimated?

**Reply**: The RDmax values are fixed in time but varying in space. They are generated from soil property maps based on the relations between soil properties and effective rooting depth described on page 25 of Refsgaard et al. (2011) (http://vandmodel.dk/xpdf/77-2011_vandbalance.pdf ). The RDmax values are scaled so that the mean values of RDmax across the entire country match the average calibrated root depths of the original DK-model.  This is done to ensure the water balances of the original and modified DK-model does not deviate too much, while avoiding a time consuming (several months of simulation time) recalibration of the modified DK-model.

Refsgaard, J. C., Stisen, S., Højberg, A. L., Olsen, M., Henriksen, H. J., Børgesen, C. D., Vejen, F., Kern-Hansen, C., and Blicher-Mathiesen, G.: DANMARKS OG GRØNLANDS GEOLOGISKE UNDERSØGELSE RAPPORT 2011/77, Geological Survey of Danmark and Greenland (GEUS), 2011.

• I don't understand what "matching the original DK model" for RD and KC means. I thought the aim is to make them variable. How did you achieve to make them matching, by parameter calibration? Please elaborate a bit more on that.

**Reply:** We agree with the reviewer that it becomes a little bit confusing. In all the study we have used the original configuration of the DK-Model as reference. To derive the LAI from NDVI, RD, etc… The aim of doing this is to keep the spatial average values of the parameters across the country as similar as possible and focus only on changing/improving the spatial distribution of them without affecting so much the mean model inputs and therefore avoiding a new model calibration. See also the reply above.

• P8L3-6: At P6L14-16: you state an actual value comparison is not anticipated. Here you are calibrating your TSEB model with eddy covariance data. Why? Please elaborate more on that.

**Reply:** We decided to adjust a few of the vegetation related parameters of the TSEB model based on land cover specific eddy covariance data from 3 different towers representing different vegetation types. Even though we are not utilizing the absolute values of the TSEB (only the pattern) the patterns are representing the differences in space and therefore we believe it is valuable compare and adjust the TSEB to different land covers where flux tower data were available.

We will elaborate on this in a revised manuscript, which will also include a clearer explanation of the sensitivity analysis and calibration of the TSEB.

• P8L1-2: Wouldn't a variance based sensitivity method better fit the purpose of identifying the parameters which have to be used for model calibration instead of the derivative based approach applied herein? Probably provide some details about the chosen sensitivity approach.

**Reply:** Yes a variance based sensitivity method would be better for the purpose of identifying parameters which have to be used for model calibration. However, our sensitivity analysis was not really designed for that purpose. We use the simple sensitivity analysis to illustrate that the TSEB is mainly sensitive to the remotely sensed variables and the climate forcing which are not subject to calibration, and less sensitive to the vegetation parameters. Subsequently we adjust a few of the TSEB vegetation parameters to get a better discrimination between land cover classes, but the selection of parameters is based more on subjective choices than the sensitivity analysis. We will explain this better in a revised manuscript and also argue why we select the parameters we do for calibration.

• P9L5,L8: Please make a distinction between the terms parameter and variable, the reader gets confused otherwise.

**Reply:** Sorry you are right, changed to**:**

*"The results show that the most sensitive variable for the estimation of AET is LST. Interpreting the sensitivity values for each group individually stress that, for the remote sensing input, parameters that are directly related to LST such as emissivity of vegetation (EmissV) and soil (EmissS) are characterized by a high sensitivity as well. The next group, forcing data, exhibited high sensitivity for all variables, except for wind speed. Overall Air temperature (TempAir) is the most sensitive forcing variable."*

• P9L8/Fig 3: better: TempA = Ta

**Reply:** Thanks for the suggestion. The legend has been modified in the figure.

• Fig. 3: LAIAgri max , LAIForest max , LAIMeadow max do not appear.

**Reply:** When the sensitivity analysis was conducted, LAI was not evaluated specifically. Instead we evaluated the sensitivity of the fraction of green vegetation fg. Later when calibrating the model we utilized the LAImax for each land cover class to adjust the fg, which is estimated based on equation 5 and 6. (we unfortunately found a typo in eq5, which also includes LAImaxclass) and linearly proportional to LAImaxclass   We realize that this part of the manuscript is weakly explained and we will improve it in a revision.

• P9L14 & P8L5-6: Why did you select only those 4 parameters out of 10. For PT the others seems to be more sensitive then the forest PT, for example.

**Reply:** The choice of parameters was subjective. We chose not to calibrate PTmeadow and PTagri, because there is not really any physical reason that it should deviate from the original value of 1.28. In contrast literature suggests that PTforest is generally lower than 1.28, and therefore we decided to calibrate that value. Canopy height was considered better parameterized with realistic values for both forest and seasonally varying crop height for agriculture, therefore we preferred not to adjust the canopy height. Leaf width was not included due to low sensitivity.

• Please justify the assumption to add the residual energy to LE. I only know approaches using corrections based on the Bowen ratio or adding the residual energy to SH.

**Reply:** We used the standard Bowen ratio corrected data from the EC systems for our calibration of TSEB. The purpose of adding the residual energy to LE or SH in the figure was mainly to illustrate the size of the energy balance closure issue and to illustrate that the TSEB estimates were generally within the limits.

• Fig. 4: Thanks for including error bars to the plot. I think it is misleading showing only the error bars of the observation. Could you also show error bars on the simulation, e.g., emerging from different parameter sets?

**Reply:** The idea of the figure is to show that values of ET that we estimated from TSEB are within the uncertainty of the reference data. The red lines in the figure represent the range of the energy balance closure problems, they do not include the entire uncertainty range, since uncertainty in the measurements themselves are not included. It would be great to have an estimate of the uncertainty of the TSEB, but even if we included uncertainty arising from the vegetation parameters that would only cover a fraction of the true uncertainty. Given that the TSEB is mainly sensitive to the LST, albedo and climate forcing, the uncertainty of those constitutes a much larger uncertainty. Therefore uncertainty bars based on parametrization alone would be misleading. We will elaborate on this in the revised manuscript.

• P9L20: Could you please mentioned the spatial resolutions of EC and RS data? • The results section is missing in general a discussion with other studies. E.g., estimating ET from MODIS data comparing to Mu et al. (2007, 2011).

**Reply:** The resolution of the remote sensing dataset is at 1 km.

EC spatial resolutions can vary. The resolution depends on the heights at which the measurements are taken. Another factor that affects the footprint size is the surface roughness, and the last is the thermal stability therefore suggesting a footprint size is difficult. We know that at all stations the instruments are located at best possible locations and heights to be representative of the area and capturing the fluxes in area of tens of meters to hundreds of meters, but under some conditions might be affected by fluxes from areas nearby but not much as the EC captures most of the eddies from the area nearby the station.

 • P10L14-15: Is it reasonable to observe lower ET for forest areas? Wouldn't canopy interception increase ET only after precipitation events?

**Reply:** We agree with the comment from the reviewer. However, we are only evaluating non-cloudy days, meaning that there is no rainfall and thereby no interception. When we compared the data from the EC of the different sites, we noticed that the ET of croplands was higher than those obtained in the forest areas, for these specific cloudfree days especially during the peak of the crop growing season (May-July). It has to be remembered that the days we are evaluating are not representative of all conditions but limited to cloudfree conditions. Most probably the AET maps for all weather conditions would look different.

• P10L17: are causing differences in area 2 in the model domain

**Reply:** We changed the text in the manuscript.

• P10L19, Fig 6, P11L5, and others: I am very sorry but I cannot observe the pronounced difference between zone 5 and 6. Could you provide some more information on that, e.g., zoomed plot numerical analysis? At the provided plots I do not see this features.

**Reply:** We agree that it is difficult to observe the difference in the contrast between domain 5 and 6. This is partly as a consequence of using the same color ramp and color stretch for all the maps in the figure. We have changed the text to focus on the differences between domains 1,2 and 3 in the DK-model.

Figure 8 below, which might be a better example of the differences between domain 5 and 6.

[Figure]

P10L10: reformulate: extracted

**Reply:** I think you mean P10L20. We have reformulated to:

Figures 5 and 6 indicate that there is very little resemblance between the spatial patterns of the TSEB ET and the DK-model simulations on the national scale.

• P10L24: ... does not necessarily lead to reasonable ET ...

**Reply:** Thanks for the suggestion. Text has been modified.

• Fig. 9 and P11L9-15: Could you provide numerical evidence to the explanatory variables of the spatial patterns of ET. I can see the E-W gradient in clay content and ET but the others are not observable. Consider rewriting or deleting some of your conclusions since they are not supported by your data. Possibilities to get evidence: scatterplots or SPEARMAN rank correlations.

**Reply:** We agree that the visual interpretation should be backed by some quantification; we will add correlation coefficients between variables and TSEB AET to the maps in figure 9.

We do not consider necessary incorporating the scatter plots in the manuscript as there are already a large number of figures and maps in it. However, we will show them here:

Density scatter plot TSEB- LAI (r =-0.15)

[Figure]

Density scatter plot TSEB- LST (r =-0.50)

[Figure]

Density scatter plot TSEB- Clay Fraction (r = 0.44)

[Figure]

• I miss the comparison of the model performance in streamflow and groundwater table between the original and modified DK model. I understand that the spatial representativeness of the modified DK has improved compared to the original one. But shouldn't be made sure that the water balance is still sufficiently represented by assessing the streamflow and groundwater tables since that is the major purpose of the model? Therefore, the model performance shouldn't deteriorate significantly if evaluated with those variables

**Reply:** We agree with the reviewer that some information on the performance of the model should be provided. We will include in the manuscript a new section that evaluates the performance of the Original and the modified version of the DK-Model with respect to streamflow and groundwater head. We first included the results regarding the discharge. (See next figure)

[Figure]

**Figure 12. Model performance statistics showing the results of the original and modified versions of the DK model. Stations have been ranked by performance and presented in the x axis as a percentage. Figure shows the Nash-Sutcliffe Efficiency (NSE), water balance error (WBE), and water balance error only for the summer period (WBESummer)**

Following the suggestion of the reviewer we have also included the results of the water heads (See next figure)

[Figure]

**Figure 13. The figure shows the error and statistics in the ground water heads estimated from the original DK model and the modified version of the DK model. Stations have been ranked by performance and presented in the x axis as a percentage.**

In both cases the performance statistics have decreased, but not in a very significant way. We expect the statistics of the modified version of the DK model to be similar to those of the original model after a new model calibration that includes also a spatial performance metric in the objective functions, however, at this moment that task is not yet feasible as it requires a new framework to carry on the calibration focusing on both, the spatial pattern performance and the temporal discharge performanceWe are starting now to develop the framework and data preparation, but is still unknown when it will be finalized as it also requires a large computing time once all the model inputs and datasets are ready. It is important to highlight that the performance of the original model is a calibration performance whereas the performance of the modified model is a validation.

We will include a new section in the text were the results of the modifications are included.

**3.3 DK model performance**

*Results showing the water balance error (WBE) for all year and summer (from xxx to xxx) as well as NSE (Nash-Sutcliffe Efficiency) are presented in figure 12. The performance statistics of the DK model has change compared to its original calibrated setup. The first noticeable thing that can be concluded is that the average water balance error changes from a slight overestimation to a moderate underestimation (Median WBE changes from -5.5 % to 5.5.%for the original and modified models respectively).. Regarding, the summer water balance which is expected to be influenced the most by the model modifications; the picture is similar although the performance get worse with a larger positive bias.  The NSE showed a decrease in performance, from NSE= 0.72 in the original DK model to NSE=0.67 in the modified version.*

*Ground water heads were also evaluated and results are shown in figure 13. The results in this case are very similar between the original version and the modified one. Statistics showed a RMSE of 5.5 m in both cases, which sounds like a large error, however, the median, is below 1 m.*

*The results of this comparison are promising considering that the model was not re-calibrated with the new inputs. In the future, the model will be recalibrated including a spatial metric as an objective function during the calibration, and it is believed that especially in model bias can be minimized*

**3 Technical corrections**

• Fig. 1: excluded in figure) - parentheses missing, consider using different symbols for Agri and Meadow because they are hard to distinguish.

[Figure]

Corrected

• P2L10: rational behind developing.  Corrected.

• P2L21-25: because you do not provide exhaustive list of references for each application example i suggest to use 'e.g.,' in front of the references. Corrected.

• P3L7: Figure 1 presents the herein used study domain. Text added to caption of figure 1. Corrected.

P4L27: I would put the LAI sentence to the previous paragraph and start the new paragraph with: "The study focuses". Corrected.

• P5L6: delete successfully after Boegh et al. Corrected.

• P5L7: this study instead of the study - you should check that in the entire manuscript

**Reply:** We checked and corrected in the manuscript.

• P5L7: similar approach was applied where ... - please delete "was applied" later in the sentence Corrected.

 • P5LL25:please do not introduce abbreviation like 10U which are never used in the manuscript Corrected.

 • P6L5: To identify the different periods, first, the dates ... Corrected.

• Is LAIi.max the same as LAIMax in Eq. 4 and Eq. 9? check consistency

**Reply:** No, LAIiMax is for the current grid cell, while LAImax (should be named LAImaxclass) is for the entire land cover class.

• P6L6: breakpoint Fig. 2 not figure 3 Corrected.

• P6L6: better: breakpoint Fig. 2, i.e., the onset of the growing season

Corrected. We have rephrased this section.

• P6L8: Eq. 6 instead of 5

Corrected

• you are switching from Eq. to equation and Fig. to figure in the entire manuscript check consistency • probably check for figure and equation referencing in the entire manuscript

Corrected. We have checked the consistency in the manuscript for EQ and Fig.

• Eq. 7 & 8: netRad and net radiation - consistency

Corrected

• I would suggest to use formula symbols like Rn instead of words like netRad

Corrected

• P6L20: The resulting maps ...

Corrected

• P6L21: in just climatological maps

Corrected

• P6L26: latent heat (LE) or evapotranspiration measurements are

Corrected

• P6L29: which is usual instead of not unusual

Corrected

• P7L7: RDi = RDi

Corrected

• P10L11: Fig. 5 instead of Fig. 56

Corrected

• P10L11: pattern identified the TSEB

Reply: I think you meant P10L20. Corrected

• P12L22: the meso.. instead of The meso..

Corrected

---

## Author Response (AR1)

Correspondence to Gorka Mendiguren (gmg@geus.dk)

Response to anonymous Referee #1

**Reply:** The authors would like to thank the reviewer for his/her detailed and elaborated review of the manuscript. The comments and suggestions are very much taken into thorough consideration as we believe they will improve the reading and add a significant contribution to the manuscript increasing the scientific quality. We are very pleased to read that he/she considers the manuscript appropriate for publication after major revision in Hydrology and Earth System Sciences (HESS). We hope that the changes conducted in the revised version of the manuscript will be well received by the reviewer and that he/she will regard the publication as fit for submission in Hydrology and Earth System Sciences.

**1 General comments**

The manuscript is dealing with the topic of spatial patterns in distributed hydrological modeling. The authors present a study in which they derived a remote-sensing based ET dataset which they use to improve the spatial patterns of the MIKE-SHE national model of Denmark (DK model). These improvements are achieved by adjusting the parameterizations and input data of the existing DK model. They conclude, that spatial patterns of remote sensing data are a valuable information to inform hydrologic models about spatial patterns, whereas these models are usually calibrated on integral signals, such as streamflow.

The topic fits the scope of HESS and is of scientific interest. The chosen methods seems to be appropriate but some clarification in the methods section is still missing. The authors introduce novelty to the field of applying remotely sensed data for hydrological modeling by consideration of bias insensitive pattern matching techniques. The manuscript lacks, here and there, the soundness of the applied evaluations using scientific methods. A lot of evaluation of the spatial patterns in figures 5 to 7 is done on visual basis without proper numerical/scientific quantification. Some features, e.g., the often mentioned "clear" distinction between model region 5 and 6 are hard to observe for the reader if even existing. Further, the adopted DK model is never evaluated regarding streamflow or groundwater levels which is the main purpose/application of this model. Another criticism is the absence of a proper discussion of the findings of this study, there is little referencing to any other study such as Mu et al. (2007, 2011) for remotely sensing based ET estimates. With exception of section 2.2 the manuscript is well written and good to understand. It could improve by better organization of the sections. I would swap sections 2.3 and 2.4 because sections 2.2 and 2.4 belong together in my opinion. Further, I suggest to fully reorganize and rewrite section 2.2 since it is hard to follow and the storyline is missing in there. Concluding, I suggest to accept the manuscript for publication in HESS after major revision.

**Reply:**

We agree with the reviewer that the inclusion of statistics of model performance regarding streamflow and groundwater head will add significantly to the manuscript. Initially we decided that the comparison between the performance of the original and modified DK-model would not be fair, since the original model has been calibrated against streamflow and head, whereas the modified has not (single model run time of around 40 hours makes re-calibration a huge task). However we have decided to include a full section comparing the streamflow statistics, water balance errors and groundwater head errors with the original and modified version of the DK model in the manuscript.

Regarding the numerical quantification of the spatial patterns similarities, this is something our group has worked extensively on (Koch et al., 2015) and we currently have another manuscript under review that addresses this issue. However, for this particular study the detailed numerical quantification of the pattern similarities were believed to be a bit irrelevant because the differences are so large between the TSEB and DK-model and therefore a visual interpretation was thought as more appropriate. However we did quantify the correlation coefficients in Figure 10. In a revision we will also add correlations between TSEB AET and different variables (fig 9) to provide a more numerical quantification (Page 12, Lines 29-30).

Regarding the description of the differences between model domains, we agree that this has not been written very clearly and this was improved during revision. But the point was that the lack of spatial consistency in model parametrization results in simulated pattern artifacts. This is still the case and is now exemplified better.

Regarding Mu et al., 2007 we are aware of that product, but chose to produce a satellite based AET dataset based on land surface temperature (LST), which is not included in the Mu model. Their model is primarily driven by LAI and NDVI. This choice is justified by the fact that in our study the LST/TSEB and NDVI/LAI patterns are sometimes very different, especially during the senescence phase. We added an elaborated description of the TSEB-method and discussed how it differs from other regional AET methods and argue why we chose the TSEB.

Koch, J., K. H. Jensen, and S. Stisen (2015), Toward a true spatial model evaluation in distributed hydrological modeling: Kappa statistics, Fuzzy theory, and EOF-analysis benchmarked by the human perception and evaluated against a modeling case study, Water Resour. Res., 51, 1225–1246, doi:10.1002/2014WR016607.

**2 Specific comments**

Introduction: The introduction is well written and gives a appropriate overview on the topic and shows the novelty of this study compared to existing research.

Methods: In general section 2.2 should be reorganized and rewritten because it is difficult to follow (I am missing the storyline here) and hard to understand what all the variables and equations are needed for. I think a major thing missing here is the presentation of the TSEB equation to assess which variables are needed in order to estimate ET. This will make clearer why you estimate LAI and vegetation height among others.

- please include TSEB equations

**Reply:** We agree that the method section should be reorganized, and this was done by clearly splitting the TSEB model and the hydrological model in two sections with the appropriate subsections on methodology, remote sensing derived inputs and calibration/validation setup.

The TSEB method cannot be summarized into a single equation, but we understand that it becomes difficult for the reader to follow the TSEB model without some extra explanation. Therefore following the reviewer recommendation we have extended the model description, included several equations that hopefully helps the reader to understand how TSEB works.

However, the model implies several steps and some theory is involved as well so the reader is encouraged to read the article from Norman et al. 1995 to get a complete idea of how the model works in detail.

The text will be modified and the description of the model gets in the manuscript will get this (Page 4 Line 4 – Page 5 Line 9:

**"2.1 TSEB setup**

**2.1.1 TSEB theory**

*The Two Source Energy Balance Model (TSEB) proposed by Norman et al. (1995) is used to retrieve mean monthly maps of ET across Denmark. In our study we have incorporated the code which is provided by the pyTSEB package (https://github.com/hectornieto/pyTSEB last accessed 30/01/2017). The applied model is a two layer model that treats soil and vegetation separately and estimates fluxes on the basis of LST and air temperature ($T_{Air}$) among other input variables. As presented in Norman et al. (1995), and presented here in a very simplified explanation. The model is based on the energy balance equation:*

$$LE[Wm^{-2}] = R_n - H - G$$
*(1)*

*Where H is the sensible heat flux, G is the ground heat flux, $R_n$ is the net radiation and LE is the latent heat flux all in $Wm^{-2}$.*

*The sensible heat flux (H) is calculated as:*

$$H[Wm^{-2}] = H_C + H_S = \rho C_P \left[ \frac{T_C - T_A}{R_A} + \frac{T_S - T_A}{R_A + R_S} \right] \qquad (2)$$

*Where $H_C$ and $H_S$ are the sensible heat flux for the canopy and soil respectively in $Wm^{-2}$, $T_C$ and $T_S$ are the canopy and soil temperatures (K), $\rho C_P$ is the volumetric heat capacity of air ($Jm^{-2}s^{-1}$), $R_S$ is the resistance to heat (s $m^{-1}$) flow in the boundary layer above the soil surface and $R_A$ is the aerodynamic resistance (s $m^{-1}$) expressed as:*

$$R_A[sm^{-1}] = \frac{\left[ ln\left( \frac{z_U - d}{z_M} \right) - \Psi_M \right]\left[ ln\left( \frac{z_T - d}{z_M} \right) - \Psi_H \right]}{0.16U}, \qquad (3)$$

*where $z_U$ and $z_T$ are the height of the wind speed (U) and air temperature in meters, $\Psi_M$ and $\Psi_H$ are the adiabatic correction factors for momentum and heat, d is the displacement height ($d \approx 0.65\ h_c$, and $h_c$ is the height of the canopy (m)), $z_M$ is the displacement height for momentum ($z_M \approx h_c/8$).*

*The model starts with an iterative process in which finds the $T_C$ which satisfies the energy balance equation. The divergence of net radiation in the canopy ( $\Delta R_n$) is used to partition sensible and latent heat fluxes using the Priestley-Taylor approximation (Priestley and Taylor, 1972) for the green part of the canopy. The transpiration is given by the next eq.*

$$LE_C[Wm^{-2}] = 1.3 f_g \frac{S}{S+\gamma} \Delta R_n, \qquad\qquad (4)$$

*Where $f_g$ represents the fraction of LAI that is green, S is the slope of the saturation vapor versus temperature curve and $\gamma$ is the psychrometric constant and where $\Delta R_n$ is calculated as:*

$$\Delta R_n = R_n - R_n exp(0.9\ ln(1 - f_c)) \qquad\qquad (5)$$

*And where $f_c$ is calculated as:*

$$f_c[-] = 1 - exp\left(\frac{-0.5 LAI}{\cos\theta}\right) \qquad\qquad (6)$$

*Where $\theta$ is the viewing angle.*

*The model iterates until the energy balance equations are satisfied for soil and canopy. The readers are referred to Norman et al. (1995) to find a fully description of the model for a more detailed explanation of how the iteration process is carried out."*

Norman, J. M., Kustas, W. P., and Humes, K. S.: Source approach for estimating soil and vegetation energy fluxes in observations of directional radiometric surface temperature, Agricultural and Forest Meteorology, 77, 263-293, http://dx.doi.org/10.1016/0168-1923(95)02265-Y, 1995.

Priestley, C. H. B., and Taylor, R. J.: On the Assessment of Surface Heat Flux and Evaporation Using Large-Scale Parameters, Monthly Weather Review, 100, 81-92, 10.1175/1520-0493(1972)100<0081:OTAOSH>2.3.CO;2, 1972.

• P4L24: Is the LAI estimate sensitive to its source satellite? So is there any difference in LAI data originating from TERRA compared to AQUA?

**Reply:** I think you mean the LST instead of LAI (P4L24). TERRA and AQUA satellites have identical characteristics regarding spectral wavelengths of the sensor they carry. Both sensors operate together with different overpass times and therefore the difference in LST is mostly due to the overpass time, not due to sensor.

Regarding the estimation of LAI the MODIS product that we used combines information from both satellites, TERRA and AQUA.

• P4L30: What does BRDF mean?

**Reply:** we apologize for the typo. We will include in the text what BRDF means: Bidirectional Reflectance Distribution Function. (P5 L20)

• Eq. 1: Please state the wavelengths for B1 and B2

**Reply:** Wavelengths have been included in the text. B1=645.5nm and B2= 856.5 nm. (P5 L24)

• clarify for what purpose LAI, albedo, VH and others are needed, I think the TSEB equation will help a lot for that

**Reply:** We have included a more detailed explanation of the TSEB that hopefully helps to clarify this question. We added a brief description in the text indicating how the albedo is used to calculate the net radiation.

Albedo was used in the study to calculate the net radiation.

• Eq. 6: please explain LAIMaxClass

**Reply:** LAIMaxClass, is the maximum LAI value for a given land use class. This value is used to scale any given LAI value for a particular grid and day relatively to the land use class it belongs to**).** We will include the explanation of the LAI MaxClass in the manuscript. (P5 L30)

• Fig. 2: probably add the growing phase as a gray box, I think the red line should be dotted outside the growing phase as it was estimated with Eq. 5, merge both legends to one, caption: probably show pixel in map (Fig. 1) - row 100 and column 84 definitely means nothing to anybody, add: LAI corresponds to the left ordinate and Fg to the right one.

**Reply:** We have followed the reviewer recommendation and added a grey box in the figure. The figure looks now like this.

[Figure]

Caption in the text has been modified following the suggestions and now looks like this:

*"Figure 2. Diagram of Fraction of Green (Fg) calculation based on the leaf area index (LAI). LAI corresponds to the left ordinate and Fg to the right one. Shadow highlights the region in time were the Eq.11 is replaced by Eq.12. Data presented corresponds to an agricultural pixel from the dataset."*

• Are the data interpolated around the breaking point or is a jump appearing in Fg? How reasonable is that?

**Reply:** The data is not interpolated in the breaking point. By the approach we implemented in the study, we aimed at detecting the beginning of the growing season by finding the point in which the LAI presented an increase of 20% compared to the pre-growing season low. This approach is taken from Cong et al., 2012. Once the date where that happened (the green-up date) was found we assume that most of the vegetation can be considered and green vegetation and therefore we considered that a rapid change in Fg, reaching values close to 0.9, were meaningful. From there the Fg values follow the evolution of the LAI and decreases as soon as the senescence of the vegetation starts and LAI decreases.

• P6L5-9: How is that approach justified? Do you have any evidence with observations or references in literature?

**Reply:** The exact approach is not found in the literature; however the 20% increase in vegetation index as green-up date is taken from Cong et al. (2012). The Fg equation itself is a simplified form of the vegetation index based method by Gutman and Ignatov (1998), where we exclude the subtraction of bare soil LAI in the equation. Guzinski et al. (2013), documented the shortcoming of Vegetation index based methods regarding estimating Fg during the greening phase, where they tend to highly underestimate Fg in contrast to the senescence phase, where VI-based methods corresponds well to field observations. We agree with the reviewer that the explanation of what we conducted to retrieve the Fg was very difficult to understand and read (in addition we also had put in a wrong equation). We have completely rewritten this section hoping to make it clearer for the reader. In principle we divide

the Fg estimation into two periods, one is the greening phase and one is the senescence and winter phase. This will allow us to get a more realistic evolution in Fg, with high values throughout the greening phase and gradually changing Fg as a function of LAI during the rest of the year. The new text says:

Page 6 Lines 9-21

*"....Fraction of Green vegetation was derived from LAI following the next equation:*

$$Fg_i = \frac{LAI_i}{LAI_{MaxClass}} \qquad\qquad (EQ\ 11)$$

*Where $Fg_i$ indicates the Fraction of green for a certain pixel i, $LAI_i$ indicates the LAI value for a pixel i and $LAI_{MaxClass}$ is the maximum LAI value for a specific land cover type. The approach is a simplified form of the vegetation index based method by Gutman and Ignatov (1998). This equation was applied to needle leaf forest land cover type.*

*For the other land cover types (crops, grasslands, deciduous forest etc.) equation 11 was modified adding another term. These land cover types show a stronger seasonality and a clear distinction between a greening phase and a senescence phase. In order to represent the strong difference in fraction of green vegetation between the period before and after senescence we introduced a different equation for the period between crop emergence and senescence, where we assigned higher values of Fg to non-needle leaf forest land covers, Figure 2. For these types of vegetation Fg will be allowed to increase rapidly just after crop emergence by substituting EQ 11 by EQ 12.*

$$Fg_i = \frac{LAI_{i,max}}{LAI_{MaxClass}} \cdot (1 - e^{(-2 \cdot LAI_i)}) \qquad\qquad (EQ\ 12)$$

*Where $LAI_{i,max}$ indicates the Maximum LAI value for a pixel i.*

*This substitution is only conducted during part of the phenological year, more specifically for the period starting at the green-up date, corresponding to the point defined by an increase in 20% in LAI compared to the winter low ($LAI_{i,Min}$)(Cong et al. 2012) and continuing until the time at which LAI reaches its maximum ($LAI_{i,Max}$) (see next Figure )."  This approach will mediate the shortcomings of the vegetation index based methods, which has been shown to underestimate fraction of green during the greening phase while corresponding well to field observations during senescence (Guzinski et al. 2013).*

[Figure]

We believe these assumptions fits well with reality, since a given LAI value before and after senescence can have quite different Fg values. During the growing season most of the plant remains green, which is quite well represented with the modification including the exponential term in the equation. After the point, where vegetation has reached its maximum seasonal development (we assume at maximum LAI), senescence starts and more non- photosynthetically active regions start to appear in the plant, what is translated in lower Fg values.

Cong, N., Piao, S., Chen, A., Wang, X., Lin, X., Chen, S., Han, S., Zhou, G., and Zhang, X.: Spring vegetation green-up date in China inferred from SPOT NDVI data: A multiple model analysis, Agricultural and Forest Meteorology, 165, 104-113, http://dx.doi.org/10.1016/j.agrformet.2012.06.009, 2012.

Gutman, G., and Ignatov, A.: The derivation of the green vegetation fraction from NOAA/AVHRR data for use in numerical weather prediction models, International Journal of Remote Sensing, 19, 1533-1543, 10.1080/014311698215333, 1998.

Guzinski, R., Anderson, M. C., Kustas, W. P., Nieto, H., and Sandholt, I.: Using a thermal-based two source energy balance model with time-differencing to estimate surface energy fluxes with day-night MODIS observations, Hydrology and Earth System Sciences, 17, 2809-2825, 2013.

• Eq. 8: I think you mean EF = ET/Rn

**Reply:** Yes we meant that, we apologize for the mistake. It has been also changed in the text. (P7 L9)

• P6L30-32: I do not understand this sentence

**Reply:** We have modified the sentence as it was confusing. In the manuscript it was written as:

*"The observed values used during TSEB evaluation are the Bowen ratio (Bowen, 1926) corrected values and the associated uncertainty estimate is the span between all error in the closure problem being assigned to the latent heat and no error in the latent heat."*

And now is written as:

Page 7 Lines 26-28

*"The evaluation of the TSEB was conducted using as reference the data of the EC systems from the 3 different land cover types that were corrected for the energy closure using the Bowen ratio approach (Bowen, 1926). The uncertainty of the energy closure issue spam between all closure errors being assigned to sensible heat for the first limit and for the second limit all error being assigned to latent heat."*

Bowen, I. S.: The ratio of heat losses by conduction and by evaporation from any water surface, Physical Review, 27, 779-787, 10.1103/PhysRev.27.779, 1926.

• Eq. 9 and 10: Why was the original RD approach based on LAI adopted to NDVI. LAI is available as seen on previous page. Could you please elaborate a bit on that?

**Reply:** We apologize for the confusion, Koch et al. 2017 mentions describing seasonal root depth evolution by scaling either LAI or NDVI, but actually applies the NDVI based scaling for Danish agricultural conditions. We have followed that implementation.

We noticed that this was not very well explained in the text and we have rewritten that part. Please, check now Page 8 Lines 20-Page 9 Line 11.

• Eq. 9: Please explain NDVIi the same as NDVImax

**Reply:** We have rewritten all this section in the new manuscript and explained the terms.

Page 8 Lines 20-Page 9 Line 11.

• P7L7: LAI in meters?

**Reply:** Sorry, only root depth in meters, LAI in [m2/m2]

• How is RDmax estimated?

**Reply**: The RDmax values are fixed in time but varying in space. For the very sandy soils in Western Denmark, the effective maximum routing depth is known to be lower than for the regions with lower sand content. This is described on page 25 of Refsgaard et al. (2011) (http://vandmodel.dk/xpdf/77-2011_vandbalance.pdf ) and in (Jensen and Jensen 1999). The RDmax values are scaled as a function of clay content so that the mean values of RDmax across the entire country match the average calibrated root

depths of the original DK-model.  This is done to ensure the water balances of the original and modified DK-model does not deviate too much, while avoiding a time consuming (several months of simulation time) recalibration of the modified DK-model. It also has to be acknowledged that the Root depth in the DK-model is considered an effective parameter, that also accounts for variations in vegetation type, phenology and soil types, because it is the main control in the ET estimation.

Jensen and Jensen (1999) Textbook - Soilphysics and agricultural meteorology (In Danish)

• I don't understand what "matching the original DK model" for RD and KC means. I thought the aim is to make them variable. How did you achieve to make them matching, by parameter calibration? Please elaborate a bit more on that.

**Reply:** We agree with the reviewer that it becomes a little bit confusing. In all the study we have used the original configuration of the DK-Model as reference. To derive the LAI from NDVI, RD, etc… The aim of doing this is to keep the spatial average values of the parameters across the country as similar as possible and focus only on changing/improving the spatial distribution of them without affecting so much the mean model inputs and therefore avoiding a new model calibration. See also the reply above.

• P8L3-6: At P6L14-16: you state an actual value comparison is not anticipated. Here you are calibrating your TSEB model with eddy covariance data. Why? Please elaborate more on that.

**Reply:** We decided to adjust a few of the vegetation related parameters of the TSEB model based on land cover specific eddy covariance data from 3 different towers representing different vegetation types. Even though we are not utilizing the absolute values of the TSEB (only the pattern) the patterns are representing the differences in space and therefore we believe it is valuable compare and adjust the TSEB to different land covers where flux tower data were available.

We will elaborate on this in a revised manuscript, which will also include a clearer explanation of the sensitivity analysis and calibration of the TSEB.

• P8L1-2: Wouldn't a variance based sensitivity method better fit the purpose of identifying the parameters which have to be used for model calibration instead of the derivative based approach applied herein? Probably provide some details about the chosen sensitivity approach.

**Reply:** Yes a variance based sensitivity method would be better for the purpose of identifying parameters which have to be used for model calibration. However, our sensitivity analysis was not really designed for that purpose. We use the simple sensitivity analysis to illustrate that the TSEB is mainly sensitive to the remotely sensed variables and the climate forcing which are not subject to calibration, and less sensitive to the vegetation parameters. Subsequently we adjust a few of the TSEB vegetation parameters to get a better discrimination between land cover classes, but the selection of parameters is based more on subjective choices than the sensitivity analysis.  We will explain this better in a revised manuscript and also argue why we select the parameters we do for calibration.

• P9L5,L8: Please make a distinction between the terms parameter and variable, the reader gets confused otherwise.

**Reply:** Sorry you are right, changed to**:**

*"The results show that the most sensitive variable for the estimation of AET is LST. Interpreting the sensitivity values for each group individually stress that, for the remote sensing input, parameters that are directly related to LST such as emissivity of vegetation (EmissV) and soil (EmissS) are characterized by a high sensitivity as well. The next group, forcing data, exhibited high sensitivity for all variables, except for wind speed. Overall Air temperature (TempAir) is the most sensitive forcing variable."*

• P9L8/Fig 3: better: TempA = Ta

**Reply:** Thanks for the suggestion. The legend has been modified in the figure.

• Fig. 3: LAIAgri max , LAIForest max , LAIMeadow max do not appear.

**Reply:** When the sensitivity analysis was conducted, LAI was not evaluated specifically. Instead we evaluated the sensitivity of the fraction of green vegetation fg. Later when calibrating the model we utilized the LAImax for each land cover class to adjust the fg, which is estimated based on equation 5(11) and 6(12). (we unfortunately found a typo in eq5, which also includes LAImaxclass) and linearly proportional to LAImaxclass   We realize that this part of the manuscript is weakly explained and we will improve it in the revision.

• P9L14 & P8L5-6: Why did you select only those 4 parameters out of 10. For PT the others seems to be more sensitive then the forest PT, for example.

**Reply:** The choice of parameters was subjective. We chose not to calibrate PTmeadow and PTagri, because there is not really any physical reason that it should deviate from the original value of 1.28. In contrast literature (Komatsu, 2005) suggests that PTforest is generally lower than 1.28, and therefore we decided to calibrate that value. Canopy height was considered better parameterized with realistic values for both forest and seasonally varying crop height for agriculture, therefore we preferred not to adjust the canopy height. Leaf width was not included due to low sensitivity.

Komatsu, H.: Forest categorization according to dry-canopy evaporation rates in the growing season: comparison of the Priestley–Taylor coefficient values from various observation sites, Hydrological Processes, 19, 3873-3896, 10.1002/hyp.5987, 2005.

• Please justify the assumption to add the residual energy to LE. I only know approaches using corrections based on the Bowen ratio or adding the residual energy to SH.

**Reply:** We used the standard Bowen ratio corrected data from the EC systems for our calibration of TSEB. The purpose of adding the residual energy to LE or SH in the figure was mainly to illustrate the size of the energy balance closure issue and to illustrate that the TSEB estimates were generally within the limits.

• Fig. 4: Thanks for including error bars to the plot. I think it is misleading showing only the error bars of the observation. Could you also show error bars on the simulation, e.g., emerging from different parameter sets?

**Reply:** The idea of the figure is to show that values of ET that we estimated from TSEB are within the uncertainty of the reference data. The red lines in the figure represent the range of the energy balance closure problems, they do not include the entire uncertainty range, since uncertainty in the measurements themselves are not included. It would be great to have an estimate of the uncertainty of the TSEB, but even if we included uncertainty arising from the vegetation parameters that would only cover a fraction of the true uncertainty. Given that the TSEB is mainly sensitive to the LST, albedo and climate forcing, the uncertainty of those constitutes a much larger uncertainty. Therefore uncertainty bars based on parametrization alone would be misleading. We will elaborate on this in the revised manuscript.

• P9L20: Could you please mentioned the spatial resolutions of EC and RS data? • The results section is missing in general a discussion with other studies. E.g., estimating ET from MODIS data comparing to Mu et al. (2007, 2011).

**Reply:** The resolution of the remote sensing dataset is at 1 km.

EC spatial resolutions can vary. The resolution depends on the heights at which the measurements are taken. Another factor that affects the footprint size is the surface roughness, and the last is the thermal stability therefore suggesting a footprint size is difficult. We know that at all stations the instruments are located at best possible locations and heights to be representative of the area and capturing the fluxes in area of tens of meters to hundreds of meters, but under some conditions might be affected by fluxes from areas nearby but not much as the EC captures most of the eddies from the area nearby the station.

 • P10L14-15: Is it reasonable to observe lower ET for forest areas? Wouldn't canopy interception increase ET only after precipitation events?

**Reply:** We agree with the comment from the reviewer. However, we are only evaluating non-cloudy days, meaning that there is no rainfall and thereby no interception in our evaluation. When we compared the data from the EC of the different sites, we noticed that the ET of croplands was higher than those obtained in the forest areas, for these specific cloud free days especially during the peak of the crop growing season (May-July). It has to be remembered that the days we are evaluating are not representative of all conditions but limited to cloud free conditions. Most probably the AET maps for all weather conditions would look different.

• P10L17: are causing differences in area 2 in the model domain

**Reply:** We changed the text in the manuscript.

• P10L19, Fig 6, P11L5, and others: I am very sorry but I cannot observe the pronounced difference between zone 5 and 6. Could you provide some more information on that, e.g., zoomed plot numerical analysis? At the provided plots I do not see this features.

**Reply:** We agree that it is difficult to observe the difference in the contrast between domain 5 and 6. This is partly as a consequence of using the same color ramp and color stretch for all the maps in the figure. We have changed the text to focus on the differences between domains 1,2 and 3 in the DK-model.

Figure 8 below, which might be a better example of the differences between domain 5 and 6.

[Figure]

P10L10: reformulate: extracted

**Reply:** I think you mean P10L20. We have reformulated to:

Figures 5 and 6 indicate that there is very little resemblance between the spatial patterns of the TSEB ET and the DK-model simulations on the national scale.

• P10L24: … does not necessarily lead to reasonable ET …

**Reply:** Thanks for the suggestion. Text has been modified.

• Fig. 9 and P11L9-15: Could you provide numerical evidence to the explanatory variables of the spatial patterns of ET. I can see the E-W gradient in clay content and ET but the others are not observable. Consider rewriting or deleting some of your conclusions since they are not supported by your data. Possibilities to get evidence: scatterplots or SPEARMAN rank correlations.

**Reply:** We agree that the visual interpretation should be backed by some quantification; we will add correlation coefficients between variables and TSEB AET to the maps in figure 9.

We do not consider necessary incorporating the scatter plots in the manuscript as there are already a large number of figures and maps in it. However, we will show them here:

Density scatter plot TSEB- LAI (r =-0.15)

[Figure]

Density scatter plot TSEB- LST (r =-0.50)

[Figure]

Density scatter plot TSEB- Clay Fraction (r = 0.44)

[Figure]

• I miss the comparison of the model performance in streamflow and groundwater table between the original and modified DK model. I understand that the spatial representativeness of the modified DK has improved compared to the original one. But shouldn't be made sure that the water balance is still sufficiently represented by assessing the streamflow and groundwater tables since that is the major purpose of the model? Therefore, the model performance shouldn't deteriorate significantly if evaluated with those variables

**Reply:** We agree with the reviewer that some information on the performance of the model should be provided. We will include in the manuscript a new section that evaluates the performance of the Original and the modified version of the DK-Model with respect to streamflow and groundwater head. We first included the results regarding the discharge. (See next figure)

[Figure]

**Figure 12. Model performance statistics showing the results of the original and modified versions of the DK model. Stations have been ranked by performance and presented in the x axis as a percentage. Figure shows the Nash-Sutcliffe Efficiency (NSE), water balance error (WBE), and water balance error only for the summer period (WBESummer)**

Following the suggestion of the reviewer we have also included the results of the water heads (See next figure)

[Figure]

**Figure 13. The figure shows the error and statistics in the ground water heads estimated from the original DK model and the modified version of the DK model. Stations have been ranked by performance and presented in the x axis as a percentage.**

In both cases the performance statistics have decreased, but not in a very significant way. We expect the statistics of the modified version of the DK model to be similar to those of the original model after a new model calibration that includes also a spatial performance metric in the objective functions, however, at this moment that task is not yet feasible as it requires a new framework to carry on the calibration focusing on both, the spatial pattern performance and the temporal discharge performance. We are starting now to develop the framework and data preparation, but is still unknown when it will be finalized as it also requires a large computing time once all the model inputs and datasets are ready. It is important to highlight that the performance of the original model is a calibration performance whereas the performance of the modified model is a validation.

We will include a new section in the text were the results of the modifications are included.

*3.3 Stream discharge and groundwater head DK model performance*

*Besides comparing the spatial patterns of the original and modified DK-model, the stream discharge and groundwater head performance is also compared. In this comparison it is important to acknowledge that the original DK- model has been calibrated against these variables, whereas the evaluation of the modified DK-model has to be considered as a validation. Results showing the annual water balance error (WBE) and summer (Jun-Jul-Aug) water balance as well as NSE (Nash-Sutcliffe Efficiency) for 181 discharge stations are presented in Figure 12. The first noticeable thing that can be concluded is that the average water balance error changes from a slight overestimation to a moderate underestimation (Median WBE changes from -5.5 % to 5.5% for the original and modified models respectively). Regarding, the summer water balance which is expected to be influenced the most by the model modifications; the picture is similar although the performance get worse with a larger positive bias. The NSE showed a decrease in performance, from NSE= 0.72 in the original DK model to NSE=0.67 in the modified version.*

*Ground water heads were also evaluated for 25,365 wells across the country and results are shown in figure 13. The results in this case are very similar between the original version and the modified one. Statistics showed a RMSE of 5.5 m in both cases, with the RMSE being dominated by relatively few very large errors while 78 % of the wells have absolute errors below 5 m. The similarity in simulated groundwater heads between the two model versions indicates that the changes in evapotranspiration patterns have little effect. However, it has to be considered that the simulated groundwater head is controlled by mainly hydraulic conductivity (which does not change between the two versions) and annual recharge upstream of the point of comparison. Since the changes in evapotranspiration patterns mainly effects the summer period, where recharge is low, the effect on annual recharge is limited. In addition, the changes in evapotranspiration patterns will redistribute recharge patterns, but the combined effect of that at some deeper well filter location will be a mixed signal causing limited changes in groundwater head.*

*The results of this comparison are promising considering that the model was not re-calibrated with the new inputs. In the future, the model will be recalibrated including a spatial metric as an objective function during the calibration, and it is believed that especially the model bias on discharge can be minimized.*

**3 Technical corrections**

• Fig. 1: excluded in figure) - parentheses missing, consider using different symbols for Agri and Meadow because they are hard to distinguish.

[Figure]

Corrected

• P2L10: rational behind developing.  Corrected.

• P2L21-25: because you do not provide exhaustive list of references for each application example i suggest to use 'e.g.,' in front of the references. Corrected.

• P3L7: Figure 1 presents the herein used study domain. Text added to caption of figure 1. Corrected.

P4L27: I would put the LAI sentence to the previous paragraph and start the new paragraph with: "The study focuses". Corrected.

• P5L6: delete successfully after Boegh et al. Corrected.

• P5L7: this study instead of the study - you should check that in the entire manuscript

**Reply:** We checked and corrected in the manuscript.

• P5L7: similar approach was applied where ... - please delete "was applied" later in the sentence Corrected.

 • P5LL25:please do not introduce abbreviation like 10U which are never used in the manuscript Corrected.

 • P6L5: To identify the different periods, first, the dates ... Corrected.

• Is LAIi.max the same as LAIMax in Eq. 4 and Eq. 9? check consistency

**Reply:** Equation 9 was unfortunately miswritten, the correct equation is in the revised manuscript

$$Fg_i = \frac{LAI_i}{LAI_{MaxClass}}$$

Regarding EQ4: LAI should be $LAI_i$ (LAI of current cell and timestep) while LAI max should be $LAI_{iMAX}$ is the maximum LAI for the current grid cell. This is different from the LAImaxclass in EQ 11 and 12, which corresponds to the maximum LAI for the entire land cover class.

• P6L6: breakpoint Fig. 2 not figure 3 Corrected.

• P6L6: better: breakpoint Fig. 2, i.e., the onset of the growing season

Corrected. We have rephrased this section.

• P6L8: Eq. 6 instead of 5

Corrected

• you are switching from Eq. to equation and Fig. to figure in the entire manuscript check consistency • probably check for figure and equation referencing in the entire manuscript

Corrected. We have checked the consistency in the manuscript for EQ and Fig.

• Eq. 7 & 8: netRad and net radiation - consistency

Corrected

• I would suggest to use formula symbols like Rn instead of words like netRad

Corrected

• P6L20: The resulting maps ...

Corrected

• P6L21: in just climatological maps

Corrected

• P6L26: latent heat (LE) or evapotranspiration measurements are

Corrected

• P6L29: which is usual instead of not unusual

Corrected

• P7L7: RDi = RDi

Corrected

• P10L11: Fig. 5 instead of Fig. 56

Corrected

• P10L11: pattern identified the TSEB

Reply: I think you meant P10L20. Corrected

• P12L22: the meso.. instead of The meso..

Corrected

Bowen, I. S.: The ratio of heat losses by conduction and by evaporation from any water surface, Physical Review, 27, 779-787, 10.1103/PhysRev.27.779, 1926.
Cong, N., Piao, S., Chen, A., Wang, X., Lin, X., Chen, S., Han, S., Zhou, G., and Zhang, X.: Spring vegetation green-up date in China inferred from SPOT NDVI data: A multiple model analysis, Agricultural and Forest Meteorology, 165, 104-113, http://dx.doi.org/10.1016/j.agrformet.2012.06.009, 2012.
Gutman, G., and Ignatov, A.: The derivation of the green vegetation fraction from NOAA/AVHRR data for use in numerical weather prediction models, International Journal of Remote Sensing, 19, 1533-1543, 10.1080/014311698215333, 1998.
Guzinski, R., Anderson, M. C., Kustas, W. P., Nieto, H., and Sandholt, I.: Using a thermal-based two source energy balance model with time-differencing to estimate surface energy fluxes with day-night MODIS observations, Hydrology and Earth System Sciences, 17, 2809-2825, 2013.
Komatsu, H.: Forest categorization according to dry-canopy evaporation rates in the growing season: comparison of the Priestley–Taylor coefficient values from various observation sites, Hydrological Processes, 19, 3873-3896, 10.1002/hyp.5987, 2005.
Norman, J. M., Kustas, W. P., and Humes, K. S.: Source approach for estimating soil and vegetation energy fluxes in observations of directional radiometric surface temperature, Agricultural and Forest Meteorology, 77, 263-293, http://dx.doi.org/10.1016/0168-1923(95)02265-Y, 1995.
Priestley, C. H. B., and Taylor, R. J.: On the Assessment of Surface Heat Flux and Evaporation Using Large-Scale Parameters, Monthly Weather Review, 100, 81-92, 10.1175/1520-0493(1972)100<0081:OTAOSH>2.3.CO;2, 1972.
Refsgaard, J. C., Stisen, S., Højberg, A. L., Olsen, M., Henriksen, H. J., Børgesen, C. D., Vejen, F., Kern-Hansen, C., and Blicher-Mathiesen, G.: DANMARKS OG GRØNLANDS GEOLOGISKE UNDERSØGELSE RAPPORT 2011/77, Geological Survey of Denmark and Greenland (GEUS), 2011.
**Reply:** The authors would like to thank the reviewer for his/her detailed and elaborated review of the manuscript. The comments and suggestions are very much taken into thorough consideration as we believe they will improve the reading and add a significant contribution to the manuscript increasing the scientific quality. We are very pleased to read that he/she considers the manuscript appropriate for publication after major revision in Hydrology and Earth System Sciences (HESS). We hope that the changes conducted in the revised version of the manuscript will be well received by the reviewer and that he/she will regard the publication as fit for submission in Hydrology and Earth System Sciences.

**Major comments:**

Mendiguren et al. studied the importance for a hydrological model simulation to reproduce similar spatial patterns as those of remote sensing data. Their modified version of the DK-model provides a simulated evaporation result that has more similar spatial features found in the remote sensing based ET. Generally, I read the paper with great interest. The paper fits very well within the stated scope of journal. I consider that the evaluation of spatial patterns of model simulation result is still quite novel and rarely done in common hydrological model practices. However, there are still some major issues to be addressed before this manuscript is being accepted for publication.

The authors present an improved model in which remote sensing derived data were used for parameterizing vegetation parameters. They claim that the improved model provides better results as it has more similar spatial patterns as those of remote sensing data. However, it seems that the benchmarking of simulation results is limited to the evaporation flux only (especially its spatial pattern). I suggest performing more evaluation and comparison to the 'original' and 'improved' model simulation results, particularly to their discharge and groundwater head results.

**Reply:** The other anonymous reviewer addressed a similar comment. In the revised version of the manuscript we have included a new section in which we present the results of the original DK model and the modified DK version**.**

In addition, I fell that the presentation, writing and structure of the paper must be upgraded.

Often, there is no clear gap/interval between paragraphs. There are paragraphs and sentences do not flow with their previous ones. These make the paper difficult to read and understand in quite a lot of places. Moreover, in the Introduction section, I hardly find any sentences related to the actual or the main objective of the study (which is to evaluate spatial patterns of a model simulation result?). I also think that the presentation and structuring of the Methods section must be improved. Furthermore, I recommend to have separated sections of Results and Discussion. Please also see some suggestions in the following minor comments.

**Reply:** We agree with the reviewer that the main objective can be better introduced, and we strive to do so in a revised manuscript. However, we have thought a lot about splitting the results and discussion and decided that it will be better to keep them together. We feel that the interpretation and discussion is better communicated along with the presentation of results. We hope the reviewer can accept this.

Minor comments:

Page 1, lines 13-15: "The main hypothesis of the study is . . . ." I suggest rephrasing this sentence. Moreover, I could not find this hypothesis in the Introduction section.

**Reply:** We agree with the reviewer that the sentence is confusing and therefore have removed it in the new version of the manuscript.

Regarding the second part of the comment, we highlighted and clarified what the objectives of the study are including also the hypothesis of the study.

Page 1, lines 26-28: Using your modified version of the DK-model, did you get any improvements on your discharge and groundwater head simulation results?

**Reply:** No, we did not get any improvement on the results of discharge and ground water head simulations (these results will be added, see reply to reviewer 1). Generally we do not expect the results to improve for discharge and heads when using the modified DK-model without re-calibration. We recently submitted another manuscript where we include a spatial metric in the calibration of a hydrological model in a smaller sub catchment. In this study the results indicated that very similar performance metrics for discharge can be achieved when the spatial component is included (indicating limited tradeoffs), but we do not expect the results to improve, but with a closer spatial pattern to the observed using remote sensing.

Page 2, lines 21-24: I suggest including some references about the applications of using satellite data for assessing groundwater as well (e.g. Rodell et al., 2009, http://dx.doi.org/10.1038/nature08238; Sutanudjaja et al., 2013, https://doi.org/10.1016/j.rse.2013.07.022; Richey et al., 2015, http://dx.doi.org/10.1002/2015WR017349). I guess that they are relevant for your study as you use the DK-model that simulates groundwater head dynamics. I am also curious how the improvement that you introduced (based on remote sensing data) affects groundwater head simulation.

**Reply:** we have included the references in the new version of the manuscript. Included in Page 2 Line 18.

Page 3, lines 1-3: Could you please elaborate more with what you meant by the "eminent risk" here? Some references will be helpful.

**Reply:** What we mean is that when a hydrological model is calibrated individually for each model grid, with independent parameter values adjusted to fit the land surface temperature at the grid level, there is a great risk of producing a parameter field that represents a physically unrealistic spatial distribution, even though it minimizes the error in each grid, because the model parameters are used to also compensate for pixel level uncertainties in the satellite product. Thus this would overestimate the

credibility of the remote sensing data. Instead we promote the idea that the model should be calibrated using relatively few global parameters linked to transfer functions which are linked to spatial distributions of basin characteristics (e.g. DEM, soil texture, vegetation, etc. ). This approach aims at identifying which parts of the model parametrization generates these differences in the spatial patterns

Page 3, lines 14-16: Could you please elaborate more with what you meant by the "diagnostic approach"?

**Reply:** By a diagnostic approach we mean that we seek to identify which parts of the model parameterization causes the differences (errors) in spatial patterns between the satellite based estimates and models and between the two models. We will rephrase that part to make it clearer to:

*"…The model evaluation is based on a diagnostic approach inspired by the study of Schuurmans et al. (2003) who utilized satellite estimates to identify conceptual model errors in a small sub basin of the MetaSWAP model in the Netherlands. This approach aims at identifying which parts of the model parametrization generates these differences in the spatial patterns."*

Page 3, lines 26-27: Neglecting biases/differences in the absolute values is a quite brave assumption. Could you please provide some justification behind it? Why?

**Reply:** We believe that the greatest value of the satellite based AET estimates is related to the spatial pattern information and not to the absolute values, in this study. Our hydrological model evaluation scheme is based on the assumption that spatial patterns are best observed by satellite measurements whereas the overall water balance is better observed by the aggregated stream discharge measurements. Therefore we chose to only evaluate the spatial pattern of the model against the TSEB monthly maps while the water balance is evaluated against stream discharge. A limitation of the TSEB data regarding water balances is that we only observe AET on specific cloud free days and therefore our evaluation of the spatial patterns is limited to the same days, which will not guarantee a reasonable evaluation of the overall water balance.

In addition, there are different sources for the net energy data behind the hydrological model and TSEB. In the case of the DK model the potential evapotranspiration is used as provided by the Danish Meteorological Institute (DMI) whilst for the TSEB the net radiation data was obtained from the ECMWF dataset. The differences between these two introduce a small bias in AET that is not the focus of our study.

Page 3, line 30: This should be "Sections 2.2 and 2.3".

**Reply:** We apologize for the mistake. This has been corrected and updated in the new version.

Section 3: Please consider to reorganize the structure (sub-sections) of Section 2. I found that it is quite difficult to understand and follow the sequence of each step of your methods.

**Reply:** We fully agree with the reviewer that the structure of the manuscript in its actual form is difficult to follow, and we apologize for the extra effort of the reviewer to follow the study.

We have decided to reorganize the structure following the reviewer recommendation. In the new manuscript we have divided the methods in three defined groups, one for the TSEB and a second one for the hydrological model and a third for the EOF, following a more logical sequence. This will also allow a more detailed theoretical description of TSEB as requested by the reviewers.

The scheme for the new methods section in manuscript looks as follows:

2.- Methods

    2.1.- TSEB

        2.1.1 TSEB theory

        2.1.2. Derived remote sensing inputs

        2.1.3 Sensitivity analysis and TSEB calibration

    2.2.- Hydrological model

        2.2.1.- Remote sensing derived hydrological model input data

    2.3.- Spatial patterns analysis: Empirical orthogonal functions

Page 4, lines 10-11: You have discharge and groundwater head measurement data. Could you please evaluate the results of your improved model to these data?

**Reply:** Reviewer 1 has pointed to this and we agree that this information should be included in the manuscript. To reduce the size of the response, please read the response give to reviewer 1 in the last comment before the technical corrections.

We include the figures for guidance.

Water balance error:

[Figure]

[Figure]

Ground Water heads:

Please, check section 3.3 in the new version of the manuscript.

Page 4, line 19: "TSEB can successfully be applied with a single LST observation, . . . " This sentence is not clear for me. What do you mean by "single LST observation"?'

**Reply:** We agree that it is a bit confusing the sentence. The TSEB separates the fluxes for soil and canopy by first calculating the temperature of the soil and the canopy from a measurement that contains both. When two observations from the same area are measured simultaneously this can be obtained easily as we have a system that is composed of two equations with two unknowns.

$T_{Rad}(\theta) = [f(\theta)T_C^n + (1 - f(\theta))T_S^n)]^{1/n}$, where $\theta$ is the observation angle, TRad is the radiometric temperature, Ts is the soil temperature, Tc is the canopy temperature and n is the power of temperature that is usually 4.

On the other hand, when a single observation is obtained it is necessary to iterate until the temperatures of soil and canopy provide a good solution to the energy balance equation.

Equation 1: Please put the reference to this equation. Moreover, it will be very helpful if you include the unit or the dimension for every variable. Example: NDVI [-]; LAI [-]; VH [unit: m]

**Reply:** NDVI is referenced (Rouse et al. 1973) few lines up in the text but we agree that the other formulas need to present the units and the references; therefore we have included in the new manuscript the formulas with the units of the variables.

Page 5, line 1: ". . . MODIS band number." Please be more specific with these "band numbers".

**Reply**: we have included in the manuscript the wavelength each of the bands corresponds to. In the case of the study the wavelengths are B1 (645.5 nm) and B2 (856.5 nm).

Equations 2 and 3: Please put the reference to these equations (e.g. as you introduced the reference for Equation 4).

**Reply**: Equation 2 has the reference from Boegh et al. (2004) two lines before the formula, and equation is the same as Equation 2 in which the coefficients have been calculated.

Page 5, line 10: Please include the dimension/unit for the parameters alpha and beta. I guess they are dimensionless.

**Reply**: LAI is measure in m2/m2, alfa is m2/m2 and betta is dimensionless. We have modified equation in the manuscript with the units.

Page 5, line 18: This sentence does not flow well with the ones before it.

**Reply:** Sorry for the mistake. We have moved it to a more appropriate location, right after the equation.

Page 5, line 21: Could you please elaborate more with what you meant by the "highest quality pixels"?

**Reply**: The MODIS MCD43B3 provides information on albedo and the quality flags are contained in MCD43B2 product. When we use the term highest quality pixels we mean that only those pixels in which the Quality Flag is good were used to generate the maps.

We have included a better explanation in the manuscript. Now the manuscript says:

"…across different years using only the pixel where the quality flag indicated the albedo was categorized as good quality of the pixel."

Page 5, line 27: I miss the explanation why you have to calculate "Fraction of Green"? For what purpose?

**Reply**: Reviewer 1 also stated the necessity to give an explanation on this. The reason why fraction of green needs to be calculated is due to the equation the TSEB is using to calculate the fluxes.

The model uses the next equation to calculate the latent heat from the green canopy:

$$LE_C = 1.3f_g \frac{S}{S + \gamma} \Delta R_n$$

Where $f_g$ represents the fraction of LAI that is green, $S$ is the slope of the saturation vapor versus temperature curve and $\gamma$ is the psychrometric constant.

In the new revised version of the manuscript we have included a more elaborated description of the model that we hope helps to understand why the different inputs are generated.  Please, find the new explanation of the model in reviewers 1  first specific comment.

Page 5, line 27 to Page 6, line 9: Please rewrite this part. I hardly follow it. And can you please justify this assumption to the reality?

**Reply:** We agree with the reviewer that the explanation of what we conducted to retrieve the Fg was very difficult to understand and read (in addition we also had put in a wrong equation). We have completely rewritten this section hoping to make it clearer for the reader. The new text says:

*"….Fraction of Green vegetation was derived from LAI following the next equation:*

$$Fg_i = \frac{LAI_i}{LAI_{MaxClass}} \qquad\qquad\qquad (EQ\ 11)$$

*Where $Fg_i$ indicates the Fraction of green for a certain pixel i, $LAI_i$ indicates the LAI value for a pixel i  and $LAI_{MaxClass}$ is the maximum LAI value for a specific land cover type. The approach is a simplified form of the vegetation index based method by Gutman and Ignatov (1998). This equation was applied to needle leaf forest land cover type.*

*For the other land cover types (crops, grasslands, deciduous forest etc.) equation 11 was modified adding another term. These land cover types show a stronger seasonality and a clear distinction between a greening phase and a senescence phase. In order to represent the strong difference in fraction of green vegetation between the period before and after senescence we introduced a different equation for the period between crop emergence and senescence, where we assigned higher values of Fg to non-needle leaf forest land covers, Figure 2.   For these types of vegetation Fg will be allowed to increase rapidly just after crop emergence by substituting EQ 11 by EQ 12.*

$$Fg_i = \frac{LAI_{i,max}}{LAI_{MaxClass}} \cdot (1 - e^{(-2 \cdot LAI_i)}) \qquad\qquad (EQ\ 12)$$

*Where $LAI_{i,max}$ indicates the Maximum LAI value for a pixel i.*

*This substitution is only conducted during part of the phenological year, more specifically for the period starting at the green-up date, corresponding to the point defined by an increase in 20% in LAI compared to the winter low ($LAI_{i,Min}$)(Cong et al. 2012) and continuing until the time at which LAI reaches its maximum ($LAI_{i,Max}$) (see next Figure ). This approach will mediate the shortcomings of the vegetation*

*index based methods, which has been shown to underestimate fraction of green during the greening phase while corresponding well to field observations during senescence (Guzinski et al. 2013)."*

[Figure]

We believe this assumption fits well with reality, since a given LAI value before and after senescence can have quite different Fg values. During the growing season most of the plant remains green, which is quite well represented with the modification including the exponential term in the equation. After the point, where vegetation has reached its maximum seasonal development (we assume at maximum LAI), senescence starts and more non- photosynthetically active regions start to appear in the plant, what is translated in lower Fg values.

Page 6, line 10: This sentence does not flow well with the ones before it. Equations 7 and 8: What is the difference between "Net Radiation" and "netRad". If they are the same, please be consistent with your variable names. Please also include the dimension/unit.

**Reply:** the reviewer is right here. We have homogenized it and include the units (W/m$^2$)

Page 6, lines 22-23: I hardly understand this sentence.

**Reply:** What we mean is that the monthly TSEB maps are calculated based on cloud free pixels on days with a high overall fraction of cloud free pixels, therefore they are not representative of all weather conditions. Moreover, when we calculate monthly mean maps from the hydrological model we only average the exact same grids and days as used to make the TSEB maps.

Page 6, lines 24-32: "Data from three eddy covariance (EC) flux towers is used as a reference to perform a sensitivity analysis and calibration of some of the vegetation parameters of the TSEB." I guess that the methods for sensitivity analysis and calibration are given in the Section 2.4? Please consider to reorganize Section 2 so that all your method steps are presented in a logical sequence.

**Reply:** we agree with the reviewer in this point and new restructure of the manuscript has been conducted in the new version of the manuscript.

Section 2.3: It makes more sense to put this Section 2.3 directly after Section 2.1.

**Reply:** We agree with the reviewer and the manuscript sections have been rearranged in a more logical way.

Page 7, lines 8-9, Equation 9: Why did you have to substitute LAI with NDVI?

**Reply:** In the first stages of the study we used the approach based on LAI but we found some problems when utilizing that equation to convert to root depth. In this study the LAI was developed based on an empirical relationship based on an exponential equation on NDVI and LAI. When converting LAI to root depth that resulted in very abrupt temporal transitions in root depths during the year. Therefore we change the root depth calculation to be a function of NDVI in order to keep the simple linear formulation in eq. 15/16 in the revised version.

Equations 11 and 12: Please put some references.

**Reply:** We cannot provide any reference for equation 11 in the old manuscript as it was created by us during the study. We noticed that the way the equations are presented is not the best as some values appear like fix coefficients and might create confusion. We have rewritten the eq. 16 and eq. 17 in the new manuscript (P9L1-L16) more clearly. References for equations that are not from this study are now also included in the new text.

Page 8, line 1: ". . . perturbation with respect to a change in model performance." What is your model performance objective function? RMSE? NSE? KGE?

**Reply:** Mean Error was selected as the objective function. The way it was conducted was finding the parameters where it was obtained minimum error between the monthly mean values of ET for each month and the 3 different sites.

 Page 8, line 5: How did you choose these four parameters?

**Reply:** The choice of parameters was subjective. We chose not to calibrate PTmeadow and PTagri, because there is not really any physical reason that it should deviate from the original value of 1.28. In contrast literature suggests that PTforest is generally lower than 1.28, and therefore we decided to calibrate that value. Canopy height was considered better parameterized with realistic values for both forest and seasonally varying crop height for agriculture, therefore we preferred not to adjust the canopy height. Leaf width was not included due to low sensitivity. See also reply to reviewer 1 regarding the use of the sensitivity analysis.

Equation 13: What is the best value of SEOF? 0? Please clarify.

**Reply:** This is a good question. Ideally the best score should be 0 which will indicate that there is a matching in the spatial pattern. Regarding the upper limit, it will depend on the spatial variability of the data; high loading values are found for cases with a distinct variability and vice versa. This makes it difficult to put the score into context or to give a physical relevancy. Also the comparison between models or catchments may be limited.

On the other hand, as 0 is the most similar spatial pattern can be used as an objective function and therefore can be used in calibration.

Section 3: I recommend to have separated sections for Results and Discussion.

**Reply:** we consider is more appropriate the way is presented in the actual form as it allows the reader to follow the story line of the manuscript. Please also see our reply to the general comments.

Section 3.1: Sensitivity analysis: Can you please explain more about how you perturb your model input data? What is their range for the maximum and minimum values of each variable?

**Reply**: The sensitivity analysis is very simple, we perturb each variable or parameter by a fixed percentage compared to the initial and evaluates the change in objective function relative to teh change in variable/parameter. For temperatures these are changes based on their $^{o}$C values.

Page 8, line 14: How did you choose these four parameters? Based on your sensitivity analysis (Fig. 3)? How?

**Reply:** We have addressed this question previously.

The choice of parameters was subjective. We chose not to calibrate PTmeadow and PTagri, because there is not really any physical reason that it should deviate from the original value of 1.28. In contrast literature suggests that PTforest is generally lower than 1.28, and therefore we decided to calibrate that value. Canopy height was considered better parameterized with realistic values for both forest and seasonally varying crop height for agriculture, therefore we preferred not to adjust the canopy height. Leaf width was not included due to low sensitivity.

Section 3.1: TSEB calibration: How realistic is your calibrated TSEB result map (including its spatial pattern)? Did you compare it to other studies (e.g. to MODIS, GLEAM, etc.)?

**Reply:** The reviewer makes a good point here and we really consider that is a task that should be addressed in a future study. We did not conduct any comparison against other models i.e MODIS, GLEAM etc… As was mentioned in the study we could use information from three different eddy covariance towers to evaluate the estimates from TSEB.

Regarding the representativeness of the spatial patterns we believe they are representative as LST is highly correlated with ET. Moreover, as shown in the sensitivity analysis the LST drives the TSEB and as shown in figure the correlation between LST map and TSEB is quite high. We believe that LST is a key indicator of evaporative state at the surface and that LST based ET algorithms are more appropriate

than purely vegetation based ones eg. MODIS (Mu et al 2007). Other models such as GLEAM are very much fusion of models and remote sensing data (e.g. including a soil moisture model and plant water stress model) and as such becomes difficult to regard as an observation. Even though TSEB is also a model it is very much driven by the key observations of LST and vegetation, making it more observation based.

Figures 5, 6 and 7: Please use the same and consistent legend (color and values) for all figures so that they can easily be compared.

**Reply:** we agree with the reviewer that the legend should be homogenized. The spatial patterns in Figures 5,6 and 7 do not represent themselves well on a common color scale. Therefore we have now normalized the maps from figures 5,6 and 7 with their own mean to focus only on the spatial pattern, as was conducted in figure 8. However that will mean that the seasonal component of the ET will disappear. It's open for discussion which presentation is better, real values (mm/day) or normalized values on a common color scale. The results and discussion were adjusted to the new maps.

Page 9, lines 25-28: "The main aim of TSEB . . . ." It seems that the sentences do not flow with their previous ones.

**Reply:** We included a break point in the text.

Figure 8: How did you normalize all three maps? What was the motivation to normalize these three maps?

**Reply:** We noticed that we did not explain in the manuscript how the maps were normalized.  The way they were normalized was by dividing the mean map by the mean value of the mean map itself.

We have included the explanation in the caption of the figure to clarify it and in the text

Figure 10: Please improve the caption for Fig. 10. What does the color mean here?

**Reply:** In these types of plots that are called density scatter plots the warmer the color the higher number of points (higher frequency). Therefore those points that are bluish indicate low frequency of points whilst the red colors indicate high frequency of points.

Page 10, line 16: MIKE-SHE = DK-model?

**Reply:** Yes, sorry for the confusion here. The text has been checked for this confusion and replaced were necessary.

Page 11, lines 1-15: Please rephrase the sentences in these lines. This seems a very important finding in your result. I guess that this result appears very dependent on your choice to use Equation 11 (Page 7, line 19). I am wondering how you derive this equation, particularly their factor 12 and constant 0.2? Can we implement this equation for other study areas, e.g. to other climate regions (e.g. tropical areas).

**Reply:** The reviewer makes a good point here. We do not believe that this equation will provide good results in other areas, mostly because in our case was specifically calibrated for the study. Basically the equation is composed of two terms, the maximum RD and a factor that scales it and distributes it especially. For the agricultural land covers we used a more elaborated approach than the one used in forest areas for example. For the first term of the equation we established an empirical relationship between the maximum RD and clay fraction for 7 different soil types based on the data from the original DK model setup. The linear fitting gave as result the constants shown in the equation.

Page 12, line 27 to Page 13, line 16: I suggest starting a new subsection for this part.

**Reply:** We agree here with the reviewer that a subsection will be helpful. We have included it in the revised manuscript(P14L13).

Others:

Figure 1: For the figure on the right, what do the numbers (1 to 6) stand for?

**Reply:** The numbers state for the DK model subdomains. The DK model runs in 7 different domains (the one corresponding not Bornholm is not included in the study here), therefore the 6 numbers and lines that divide the country aim at showing the domain numbers and the location of each domain.

Figure 2: Please indicate the pixel (row 100, column 84) in Fig. 1.

**Reply:** We decided to make it more general and just specify the type of land cover that is represented as we also believe that does not add any additional information to the manuscript.

[revised manuscript text omitted]

---

## Author Response (AR2)

Reply to review comments from Anonymous Referee #2 and Anonymous Referee #3:

Anonymous Referee #2

We are very pleased with the comments given by the reviewers, and have addressed them below.

- Figure 9. The unit of LAI is m2/m2.

We have corrected the legend of the figure.

[Figure]

Anonymous Referee #3

We are very pleased with the comments given by the reviewers, and have addressed them below. Especially we agree that the theoretical description of the TSEB model iteration scheme was not detailed enough, and we are happy to be given the change to improve this.

Summary:

The paper presents an evaluation of the spatial pattern of distributed hydrologic model simulation (ET in this paper) against independent ET map derived from remote sensing data. The method to quantify the spatial pattern matching uses EOF. The paper updated ET parameterization of their hydrologic model based on remote sensed vegetation data to overcome issues of previous parametrization i.e., spatial discontinuity of parameter fields that are derived from per-region calibration. The paper show that updated parameterization improves ET spatial pattern over use of the previous model parameters.

Comments

The paper points out an important aspect of hydrologic model evaluation – spatial distribution of model simulations. Overall introduction nicely lays out this problem. I agree that few studies have looked at this aspect of evaluation. The update of the ET parameters for the DK model using remote sensing data is somewhat specific to their model, but I think the paper provides a good case to use remote sensing data in a conjunction with transfer functions to compute spatially distributed parameter values. I would recommend minor revision before being accepted for HESS publication. Most of my comments could be addressed by further revising texts.

Section1. One minor comment is on P3, L10-11. Is this really the most important goal of this paper? I understood that TSEB ET map is one method to generate ET maps that can be used for evaluation but potentially other ET products could have been used for this study. A primary goal of this paper is to evaluate spatial pattern of distributed ET from hydrologic model and improve the ET parameterization. If so, revise the statement of this paper goal.

Reply: We agree, this has been rephrased

Section 2.1- TSEB setup, especially theory part is a little confusing to me. It was perfectly fine till L28, and then rest of the descriptions give me hard time understand how latent heat is actually computed. There is one unknown Tc (canopy temperature). So I assume the code iterates model simulation by changing Tc, but why does this need Priestley-Taylor? Need some improved descriptions for this model iteration and how final ET is estimated. A few more minor comments include - 1) what is the assumption for ground heat transfer? And 2) it could be helpful to include a table showing all the input data (or maybe Figure 3 list complete parameter/variables?).

Reply: We agree, that the TSEB iteration approach was not properly explained. This has been elaborated in the revised manuscript. The revised TSEB theory outlines the major equations enabling the reader to follow the principle.

The TSEB model operates as a two-source model, canopy and soil and there is essentially two unknowns Tc and Ts, soil and canopy temperatures, which are related to the observed directional LST by:

$$LST(\theta)[K] \approx \left( f(\theta)T_C^4 + (1 - f(\theta)T_S^4) \right)^{1/4}$$

The model initially assumes potential evapotranspiration for the canopy fraction, the Priestly-taylor approximation; $LE_C[Wm^{-2}] = \alpha_{PT}f_g\frac{S}{S+\gamma}R_{nC}$., with $\alpha_{PT}$ = 1.26. Based on this Hc and Tc is calculated. From there Ts, Hs and LEs are estimated under the initial assumption. However, if LEs < 0 is encountered, the energy balance is not satisfied and the canopy is assumed to be stressed, meaning that the initial assumption does not hold. Subsequently, $\alpha_{PT}$ is iteratively reduced, and a new Hc->Tc->Ts->Hs->LEs is calculated, until solutions with LEs > 0 are obtained. The model iteration continues until the energy balance equations are satisfied for both soil and canopy.

In the revised manuscript we have explained this better and also listed the remote sensing and climate input data required to run the model.

Section 2.1.2 describe great deal of LAI, Green Fraction (GF). I wonder why authors did not use MODIS LAI GF product, which are widely used for the other studies.

Reply: We did initially process the MODIS LAI product in order to use it for the TSEB. However, we found that we could not get enough data coverage with highest quality flag to produce time series of LAI over Denmark in the winter period. In contrast the MODIS NDVI product higher data coverage. Therefore we used a conversion of MODIS NDVI to LAI as described in the manuscript. Following this, we believed it would be most consistent to base our Fg estimate on the same LAI data as used in the TSEB. Secondly, as described in the manuscript, we introduced a modification to traditional vegetation index based Fg-equations that minimizes the issues of underestimation of Fg in the beginning of the growing season (Guzinski et. al 2015)

Section 2.2. P9 L12-16 discuss crop coefficient (Kc). I understood that the model use some lookup table to get reference ET for a given land cover and use this Kc to scale to get ET. I would suggest describing reference ET. If readers (including myself) are not familiar with Mike-SHE structure, just need to guess it.

Reply: We rephrased to: "In addition, the crop coefficient (Kc), which is a correction factor for the reference evapotranspiration (ETref) was recalculated. The ETref, describes the climatologically based actual evapotranspiration for the reference crop (a short grass without water stress) and is here provided at a coarse spatial resolution of 20 km. The Kc-value accounts for the difference between a given crop or land surface and the reference crop by scaling the ETref to the potential evapotranspiration used in the hydrological model. In the original DK-model setup, Kc based on

lookup tables for different land covers.  In the modified parameterisation, Kc is derived from remotely sensed LAI using the approach presented in Allen et al. (1998) and used by Stisen et al. (2008):

$$Kc[-]= Kc_{(c,min)}+(Kc_{(c,max)}-Kc_{(c,min)})\cdot(〚1-e〛^{((-0.7\cdot LAI))})=0.95+0.2*(〚1-e〛^{((-0.7*LAI))})$$
(17)

Where the Kc_min and Kc_max are set to 0.95 and 1.15 respctively."

Section 2.3. I feel that Joint EOFs need descriptions on how two matrices (observation and simulation) are combined. Joining matrices in row (time) or column (space)? It would be helpful to describe how to interpret EOFs with combined matrix (like description in EOF section in Koch et al. 2015). Equation 18 is not shown in the results. I am not sure if this equation is needed.

Reply: In general, the EOF analysis can be used to study modes of variability in a dataset; either focusing on the temporal or the spatial variability. This has implications on how the data matrix should be prepared. The first is based on the spatial anomalies of the dataset (removing the mean in space from each timestep) and the latter on the temporal anomalies (removing the mean in time from each location). Let's assume that the data matrix used by the EOF analysis has dimensions m (rows) and n (columns). In our analysis we like to study the spatial patterns and therefore compute the spatial anomalies for our analysis. This is done for both matrices (obs and sim). The matrices should be organised in a way so that m is equal to the number of cells and n reflects the number of timesteps. These two matrices are then stacked to one integral matrix. They are concatenated along the temporal axis. In our case this results in a matrix with 38188 rows (cells) and 2*72 columns (72 months from sim and obs). The EOF analysis is based on eigenvalue decomposition which yields a matrix containing the eigenvectors (n*n) and eigenvalues (n*n). In our manuscript we refer to the eigenvectors as loadings which are used to compute the EOF based similarity score (equation 18). We will include essential parts of this extended explanation in section 2.3 and 3.2. We believe that it will help the read to understand the EOF methodology better.

The results of equation 18 are briefly mentioned in the results section: "The overall similarity scores derived by the EOF analysis presented the maximum value for a pattern comparison in June (≈0.11) and the minimum corresponded to a day in April (≈0.02)". Therefore we will keep equation 18 (EQ 20 now) in the revised manuscript.

Section 3.2. ET maps (Figures 5-9) look nice and support the descriptions in the result section. Somehow I expected that EOF analysis look at two comparison -1) original model's ET and TSEB ET, 2) modified model's ET and TSEB ET. What is justification of comparing two models ET instead of model versus independent dataset (i.e. TSEB ET).

Reply: We agree that comparing the DK-models with TSEB could be relevant for the analysis. However, at the current stage we were most interested in the comparison between the two model configurations in order to document the effect of the remote sensing based parameterization. Visually we can see that

the spatial patterns simulated by the modified DK-model resemble some of the dynamics found in the TSEB, but there is still space for lots of improvements which certainly require a re-calibration. Therefore we will leave the elaborated statistical spatial pattern comparison of DK-Model and TSEB to future work where we intend to recalibrate the model focusing of the ET patterns. This will include a flexible parameterization of root depth and LAI to allow the simulated spatial patterns of ET to adjust to the ones found in TSEB.

Section 3.3. First of all, does "WBE" mean runoff volume bias (against observed discharge), or literally mean long-term water balance closure error (P-RO-ET should be close to zero for long term mean)? I understood the former. If not, I wonder what is going on the model water balance (water balance error is large).
Second, this paper modifies the parameterization related to ET, therefore I would expect runoff volume is affected (that is why I thought WBE was runoff volume bias). Although NSE is the first metric to look at, maybe look at NSE decomposed metrics (bias, variability, and correlation) – see KGE paper (http://dx.doi.org/10.1016/j.jhydrol.2009.08.003). I would expect that bias is mostly affected, which contributed to NSE deterioration. This way, you could discuss more why (or how) NSE is deteriorated.

Reply: The WBE is the mean runoff volume bias compared to observed discharge. There is no model water balance error, Precip-runoff-AET-subsurface boundary flow-storage change = 0.

Secondly, the bias (our WBE) is affected as illustrated in Figure 12, both for annual biases and summer biases. Regarding the decomposition of KGE, that could be an idea, however we have focused on the metrics that are used in the formal evaluation of the DK-model, NSE, WBE(bias) and WBEsummer(summer bias) and as such the biases correspond to the bias of KGE and NSE reflect the variability and correlation (and bias).

We have pointed out that the WBE is the runoff volume bias in the revised manuscript section 3.3.

Line by line textual comments

P5, L12. Spell out VZA (I think it appears at the first time).

Done

P6, L11. 46 daily ET MAP?

Reply: I don't understand the comment. I don't find any error in the line "In order to further reduce noise, mean albedo maps are generated by creating 46 mean maps (at 8 day intervals) using all the scenes available for each 8-day interval across different years"

P7, L14. This is the first time that study period is mentioned. Maybe specify beginning of Section 2.
Done

P7, L7 and L10. Here notation is confusing because Rn is used for both instantaneous and daily. Also actual -> instantaneous?

Corrected to :

dET[$Wm{-}2$]=EF · dRn

It is further specified that for Eq 14 it is the instantaneous ET and Rn.

P7, L28. What is spam here?

Corrected to : span

P10, L14. AET. Is this actual ET? This paper does not mention "potential" ET, so just use ET?

Reply: We prefer to specifically refer to it as Actual ET to avoid confusion. Also in the revised manuscript we now refer to Potential Evapotranspiration as the Kc*ETref, see above reply.

P11, L15-18. This paragraph seems to be out of place, and not necessary.

Reply: We disagree, we believe it is a very important aspect of the evaluation approach that we focus on the consistent spatial patterns rather than the temporal dynamics of RS derived AET estimates. We would really like to keep this paragraph.

P13, L1-4. I think that discussion on this scatter plots fit better before discussing clay, LST and LAI maps (Figure 9).

Agree, the text and order of Figures 9 and 10 are reversed.

P13, L31. RS->remote sensing?

Done

[revised manuscript text omitted]